# Calibrating Decision Robustness via Inverse Conformal Risk Control

**Wenbin Zhou** [1 2]  **Shixiang Zhu** [1]

## Abstract

Robust optimization safeguards decisions against uncertainty by optimizing against worst-case scenarios, yet their effectiveness hinges on a pre-specified robustness level that is often chosen ad hoc, leading to either insufficient protection or overly conservative and costly solutions. Recent approaches using conformal prediction construct data-driven uncertainty sets with finite-sample coverage guarantees, but they still fix coverage targets a priori and offer little guidance for selecting robustness levels. We propose a new framework that provides distribution-free, finite-sample guarantees on both miscoverage and regret for any family of robust predict-then-optimize policies. Our method constructs valid estimators that trace out the miscoverage–regret Pareto frontier, enabling decision-makers to reliably evaluate and calibrate robustness levels according to their cost–risk preferences. The framework is simple to implement, broadly applicable across classical optimization formulations, and achieves sharper finite-sample performance. This paper offers a principled data-driven methodology for guiding robustness selection and empowers practitioners to balance robustness and conservativeness in high-stakes decision-making.

## 1. Introduction

Decision-making under uncertainty is a fundamental challenge in operations research, machine learning, and economics. From utilities preparing for extreme weather (Chen et al., 2025) to financial institutions managing investment risk (Fabozzi et al., 2007), decision-makers must act without knowing the exact outcomes of their choices. Robust

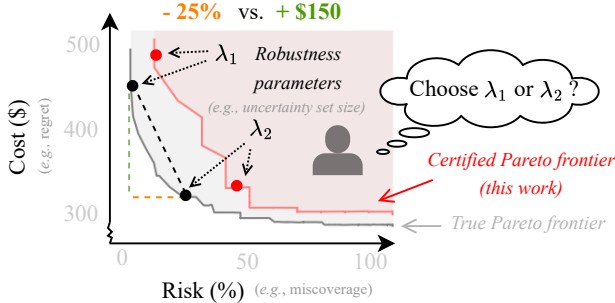

*Figure 1.* A stylized example illustrating the use of our proposed framework. Instead of heuristically fixing a risk level (*e.g.*, 5%), the decision-maker can use this frontier to select a preferred trade-off. For instance, the curve indicates that reducing risk by up to 25% (orange dashed line) yields a cost reduction of at least $150, corresponding to roughly a 30% improvement (green dashed line).

optimization (RO) has emerged as a leading paradigm for addressing this challenge: rather than optimizing for the most likely outcome, RO seeks decisions that remain effective under the worst-case realization within a prespecified uncertainty set (Beyer & Sendhoff, 2007). For instance, grid operators may schedule generator dispatch robust to worst-case demand fluctuations (Bertsimas et al., 2012), while portfolio managers may hedge against worst-case returns consistent with historical data (Goldfarb & Iyengar, 2003).

A standard way to operationalize the RO problems is through the predict-then-optimize (PTO) framework (Bertsimas & Kallus, 2020): a predictive model first suggests an uncertainty set for the outcome, and the decision-maker then chooses a decision that minimizes the worst-case loss within that set. In traditional RO, this uncertainty set is pre-specified, often as a polyhedral or ellipsoidal region around forecasts (Gabrel et al., 2014). Its size is governed by a robustness parameter, typically selected through heuristics. For instance, setting the radius to a multiple of the forecast variance. More recently, conformal prediction (CP) has emerged as a data-driven alternative for defining uncertainty sets, providing finite-sample coverage guarantees that the realized outcome lies within the set with a prescribed probability (Vovk et al., 2005). This approach has also been incorporated into robust optimization, where CP-based sets define the uncertainty region used for decision-making (Patel et al., 2024b).

[1]Heinz College of Information Systems and Public Policy, Carnegie Mellon University, Pittsburgh, PA, USA [2]Machine Learning Department, School of Computer Science, Carnegie Mellon University, Pittsburgh, PA, USA. Correspondence to: Shixiang Zhu <shixianz@andrew.cmu.edu>.

*Proceedings of the 43rd International Conference on Machine Learning*, Seoul, South Korea. PMLR 306, 2026. Copyright 2026 by the author(s).

While these approaches have expanded the scope and rigor of robust decision-making, they leave a critical question unresolved: *how should one choose the robustness level?* In RO, the radius of the uncertainty set—or in CP, the confidence level—directly governs the trade-off between protection and efficiency. In practice, this choice is typically made heuristically: by fixing conventional confidence levels (*e.g.*, 95%) or by tuning against historical simulations (Lam & Zhou, 2017). Such methods provide little theoretical guidance, and when data are scarce, simulation-based estimates can be unreliable. This gap makes it difficult for practitioners to balance safety with performance, and decisions are often more conservative than necessary.

In this paper, we propose a new statistical framework that enables decision-makers to evaluate and select robustness levels with explicit trade-offs. Our method is inspired by conformal risk control (CRC) (Angelopoulos et al., 2022), but inverts its logic. Standard CRC begins by prescribing a target risk level and adaptively constructs a set or decision rule to satisfy that risk constraint. By contrast, our inverse CRC framework takes an uncertainty set as given, and then certifies the actual risk level it incurs. This inversion is crucial: it allows us to quantify both miscoverage (the probability that the outcome falls outside the set) and regret (the expected cost relative to an oracle) across all risk levels with rigorous guarantees. Practically, this enables decision-makers to explore the trade-off between robustness and conservativeness reliably. For example, in the stylized setting of Figure 1, our method may show that reducing coverage by up to 25% can yield a cost reduction of at least $150. Such insights enable decision-makers to adjust robustness levels adaptively, achieving more cost-effective solutions rather than relying on ad hoc or overly conservative heuristics.

The approach is also straightforward to implement: given calibration data, we construct valid estimators for miscoverage and regret across all robustness levels and use them to trace out a Pareto frontier of risk trade-offs under a mild condition. If the robustness level is chosen post hoc by a human decision-maker based on the estimated trade-offs, we explicitly account for this selection by performing a final recalibration on a split dataset, which restores finite-sample validity for the reported miscoverage and regret guarantees. Theoretical results show that these estimators are reliable, enjoy sharp finite-sample guarantees, and apply broadly to various robust optimization problems. Our numerical results demonstrate that the estimator successfully traces Pareto frontiers that are both valid and tight, and can also effectively guide decision-makers toward robustness parameters that are near-optimal for achieving the desired miscoverage–regret tradeoff.

The contributions of this paper are threefold:

1. We propose a principled framework for "inverse" conformal risk control, enabling risk quantification with respect to any arbitrary loss function.
2. We develop a distribution-free estimator of both miscoverage and regret across robustness levels in robust optimization. Our approach is computationally efficient, valid in finite samples, and establishes the first certified Pareto frontier for robust optimization under mild assumptions.
3. We illustrate the broad applicability of our approach on classical optimization problems, demonstrating how it enables decision-makers to reliably calibrate robustness by balancing robustness against conservativeness.

**Related Work**   Our work is related to the predict-then-optimize (PTO) literature, which studies decision-making under uncertainty via two-stage pipelines or decision-focused learning (Bertsimas & Kallus, 2020; Mandi et al., 2024). Within this paradigm, robust optimization (RO) and distributionally robust optimization (DRO) address uncertainty by optimizing against deterministic or distributional ambiguity sets (Ben-Tal & Nemirovski, 2002; Beyer & Sendhoff, 2007; Gabrel et al., 2014; Rahimian & Mehrotra, 2019; Lin et al., 2022). Complementary to these frameworks, we adopt a diagnostic viewpoint: rather than prescribing a particular robust decision, our focus is on characterizing and certifying the risk trade-offs induced by different robustness levels, thereby supporting decision-making through principled auditing and calibration. This viewpoint also connects our work to Pareto-front analysis in multi-objective optimization (Marler & Arora, 2004; Gunantara, 2018), where trade-offs are typically explored via parameter enumeration. Building on this literature, we construct a *certified* Pareto front with explicit finite-sample guarantees using conformal prediction—an element that, to our knowledge, has not yet been incorporated into Pareto-front analysis.

Conformal prediction (CP) provides a general framework for uncertainty quantification with finite-sample validity guarantees (Papadopoulos et al., 2002; Vovk et al., 2005). Recent work has focused on improving the efficiency of CP in multi-dimensional settings by adapting prediction sets to richer geometric structures, including ellipsoidal sets (Messoudi et al., 2022; Xu et al., 2024), convex template-based sets (Tumu et al., 2024), and density-adaptive constructions (Izbicki et al., 2022; Wang et al., 2023; Messoudi et al., 2021; Sun & Yu, 2023). These contributions have laid important groundwork for a growing body of research integrating CP with robust optimization (RO), where CP is used to construct data-driven uncertainty sets for decision-making (Johnstone & Cox, 2021). Representative recent works include the use of probabilistic CP in predict-then-optimize pipelines (Patel et al., 2024b), CP-based robust control for linear–quadratic regulator systems (Patel et al., 2024a), decision-focused frameworks that leverage CP to

guarantee robustness (Cortes-Gomez et al., 2024), and multi-layer robust optimization formulations for grid operations informed by CP (Chen et al., 2025). Our work is complementary to this line of research. Rather than using CP to construct the uncertainty set, we consider settings in which the uncertainty set family is specified *a priori*, with CP used to certify risk trade-offs as the set size varies.

Conformal risk control (CRC) generalizes conformal prediction to construct loss functions with prespecified miscoverage guarantee (Angelopoulos et al., 2022). Subsequent work has substantially extended the CRC framework in multiple directions: (Farinhas et al., 2023) studies CRC under non-exchangeable data; (Blot et al., 2025) proposes an adaptive CRC procedure inspired by adaptive conformal prediction (Zaffran et al., 2022) to better exploit conditional information; (Teneggi et al., 2023) extends CRC to multi-dimensional outputs with applications to image generation; (Cohen et al., 2024) introduces a cross-validation-based approach to mitigate data scarcity; (Yeh et al., 2025) generalizes CRC to optimization problems with control over tail-sensitive risks; and (Luo & Zhou, 2025) applies CRC to image segmentation tasks. In contrast, our work addresses the inverse problem of CRC: *given a fixed loss function, how can we estimate its miscoverage?* This inverse perspective can be viewed as a generalization of inverse conformal prediction (Prinster et al., 2022; Singh et al., 2024; Zhou et al., 2025; Gauthier et al., 2025a) to CRC.

Finally, our work relates to post-hoc selection bias arising from data-dependent parameter selection (Cox, 1975). Recent works have proposed stability-based approaches (Zrnic & Jordan, 2023; Hegazy et al., 2025) and e-value-based methods in CP settings (Balinsky & Balinsky, 2024; Gauthier et al., 2025b;a). In contrast, our work follows a more typical data-splitting strategy (Cox, 1975; Wasserman & Roeder, 2009; Fithian et al., 2014; Rasines & Young, 2023).

## 2. Problem Setup

We consider a general decision-making problem under uncertainty (Bertsimas & Kallus, 2020). A decision-maker observes a contextual feature vector $X \in \mathcal{X}$ and must choose a decision variable $z \in \mathcal{Z}$ before the outcome $Y \in \mathcal{Y}$ is realized. The pair $(X, Y)$ is drawn from an underlying distribution $\mathcal{P}$. The quality of a decision $z$ in the presence of outcome $y$ is measured by a cost function $f : \mathcal{Y} \times \mathcal{Z} \to \mathbb{R}$.

To account for uncertainty from the unknown joint distribution $\mathcal{P}$, a standard approach is to introduce an *uncertainty set* that safeguards against plausible realizations of $Y$. Specifically, we introduce a family of context-dependent uncertainty sets:

$$\left\{ \mathcal{U}_\lambda(X) \subseteq \mathcal{Y} : \lambda \in \Lambda \subseteq \mathbb{R}_+ \right\},$$

indexed by a robustness parameter $\lambda$, with the convention that larger $\lambda$ yields larger sets: $\lambda_1 \leq \lambda_2 \Rightarrow \mathcal{U}_{\lambda_1}(X) \subseteq \mathcal{U}_{\lambda_2}(X)$ for all $X$. For instance, one may construct $\mathcal{U}_\lambda(X)$ as an euclidean ball centered at the conditional mean $\hat{g}(X) \approx \mathbb{E}[Y|X]$, where $\hat{g}$ is estimated from data, and take $\lambda$ as the radius of the ball. Given $X$, the robust decision is obtained by solving:

$$z_\lambda^*(X) := \arg \min_{z \in \mathcal{Z}} \max_{y \in \mathcal{U}_\lambda(X)} f(y, z). \quad (1)$$

A key challenge is to select $\lambda$ so that the robust decision strikes the right balance between *regret* and *miscoverage* as illustrated by Figure 2: If the uncertainty set $\mathcal{U}_\lambda(X)$ is too small, the decision may fail to provide sufficient protection against adverse outcomes; If the set is too large, the solution to (1) may be overly conservative and incur an undesirably large regret. We assess each $\lambda$ using two performance metrics given a pair of $(X, Y)$:

$$I_\lambda(X, Y) := \mathbb{1}\left[ Y \notin \mathcal{U}_\lambda(X) \right],$$
$$R_\lambda(X, Y) := f(Y, z_\lambda^*(X)) - \min_{z \in \mathcal{Z}} f(Y, z).$$

Specifically, for a given pair $(X, Y)$, $I_\lambda \in \{0, 1\}$ indicates whether the realized outcome $Y$ is excluded from the uncertainty set $\mathcal{U}_\lambda(X)$, and $R_\lambda$ measures the performance gap of adopting robust decision $z_\lambda^*(X)$ relative to the oracle decision under the realized $Y$.

The objective of this study is to answer two fundamental questions: For any prescribed robustness level $\lambda$, *how robust does it protect against unknown outcomes?* and *how costly is this protection in terms of conservativeness?* Formally, we seek to construct estimators $\hat{\alpha}_I(\lambda)$ and $\hat{\alpha}_R(\lambda)$ that can be trusted to provide reliable assessments for any prespecified $\lambda \in \Lambda$:

$$\hat{\alpha}_I(\lambda) \geq \mathbb{E}\left[ I_\lambda(X, Y) \right] = 1 - \mathbb{P}\left[ Y \in \mathcal{U}_\lambda(X) \right], \quad (2)$$
$$\hat{\alpha}_R(\lambda) \geq \mathbb{E}\left[ R_\lambda(X, Y) \right]. \quad (3)$$

Estimating these two upper bounds traces out the certified Pareto frontier that trades off between regret and miscoverage in the worst-case scenario, revealing how reducing miscoverage typically increases regret, and vice versa. By examining $\hat{\alpha}_I(\lambda)$ and $\hat{\alpha}_R(\lambda)$ across candidate $\lambda$, practitioners can select the point that best balances protection against conservativeness in line with their risk tolerance.

## 3. Methodology

This section first introduces a general statistical framework for "inverse" conformal risk control motivated by (Angelopoulos et al., 2022; Patel et al., 2024b), designed to estimate an upper bound on risk defined with respect to arbitrary loss functions, and construct a Pareto frontier over both

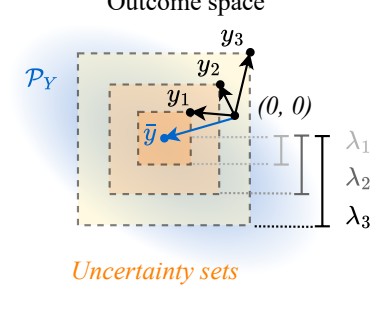

The miscoverage-regret tradeoff    Feasible region    Outcome space

(a)    (b)    (c)

*Figure 2.* Illustration of the problem setup with a simple robust linear optimization problem. Panel (a) shows the miscoverage-regret tradeoff across three robustness parameters $(\lambda_1, \lambda_2, \lambda_3)$. Panel (b) depicts the corresponding robust decisions $(z_1, z_2, z_3)$, which differ from the expected optimal decision $z^*$. Panel (c) illustrates an $\ell_\infty$-ball uncertainty set $\mathcal{U}_\lambda$ for the outcome variable $Y$, with the three adversarial outcome vectors $(y_1, y_2, y_3)$ labeled. The robustness parameters correspond to the radii of these $\ell_\infty$-norm sets.

miscoverage rate (2) and regret (3) of a robust solution to problem (1). Then, we show that these estimates are reliable and valid, *i.e.*, the expected upper bound is guaranteed to hold, and that they directly characterize the certified Pareto frontier as the robustness level $\lambda$ varies. Finally, we extend our algorithm to retain validity when used for recalibration after selection by adopting a data splitting technique.

### 3.1. Proposed Algorithm

For any prespecified $\lambda \in \Lambda$, let $\ell_\lambda : \mathcal{X} \times \mathcal{Y} \to \mathbb{R}$ be a family of nonnegative loss (*e.g.*, miscoverage or regret). We assume access to a calibration dataset of $n$ paired samples, denoted as $\mathcal{D} = \{(X_i, Y_i)\}_{i=1}^n$. Given a test data pair $(X_{n+1}, Y_{n+1})$, our goal is to construct an estimator, denoted $\hat{\alpha}_\ell(\lambda)$, from the calibration data $\mathcal{D}$ such that

$$\hat{\alpha}_\ell(\lambda) \geq \mathbb{E}\left[\ell_\lambda(X_{n+1}, Y_{n+1})\right]. \quad (4)$$

This objective subsumes both (2) and (3), and can be interpreted as an "inverse" form of the conformal risk control (CRC) (Angelopoulos et al., 2022). Specifically, CRC begins by prescribing a desired risk level $\alpha$ on the expected loss and then finds $\lambda$ that guarantees:

$$\mathbb{E}\left[\ell_\lambda(X_{n+1}, Y_{n+1})\right] \leq \alpha.$$

By contrast, our framework reverses this perspective. Instead of taking a target risk level $\alpha$ as given, we take the uncertainty set—indexed by robustness level $\lambda$—as fixed, and then ask: *what is the actual risk level guaranteed by this choice?*

Before describing our proposed algorithm, we first state two assumptions that are considered. First, we assume the exchangeability between this dataset and the test data pair.

**Assumption 3.1.** The calibration data $\mathcal{D}$ and the test point $(X_{n+1}, Y_{n+1})$ are drawn exchangeably from distribution $\mathcal{P}$.

Exchangeability is standard in the conformal prediction literature (Vovk et al., 2005) and subsumes many common settings, including *i.i.d.*. Next, assume that there exists an essential supremum to the loss function.

**Assumption 3.2.** There exists a constant $B \geq 0$ such that, for all $\lambda \in \Lambda$, $\ell_\lambda(X, Y) \in [0, B]$ almost surely.

This assumption inherits from (Angelopoulos et al., 2022), upon which our framework is built. The constant $B$ is trivially known in many settings. For example, for miscoverage losses $I_\lambda \in \{0, 1\}$, there is $B = 1$. Even in practical settings where $B$ is unknown, one may explicitly derive it using prior knowledge of the data distribution, or approximate it using data-driven estimates such as the sample maximum.

Given the two assumptions above, we propose a conformalized risk estimator as follows:

**Definition 3.3.** Let $\bar{\ell}_n(\lambda)$ denote the average calibration loss, *i.e.*, $\bar{\ell}_n(\lambda) := \frac{1}{n} \sum_{i=1}^n \ell_\lambda(x_i, y_i)$. The proposed risk estimator $\hat{\alpha}_\ell(\lambda)$ is defined as

$$\inf_{\alpha \in (0, B)} \left\{ \alpha : B - \bar{\ell}_n(\lambda) \geq \frac{\lceil (n+1)(B-\alpha) \rceil}{n} \right\}, \quad (5)$$

An upper bound to this estimator can be further derived as

$$\tilde{\alpha}_\ell(\lambda) = \frac{n}{n+1} \bar{\ell}_n(\lambda) + \frac{B}{n+1}, \quad (6)$$

which can be obtained by relaxing the ceiling operator in (5). This form requires only the empirical average calibration loss $\bar{\ell}_n(\lambda)$ plus an adjustment $B/(n+1)$ that accounts for the worst-case contribution of the unseen test point, guaranteed to lie in $[0, B]$ by Assumption 3.2. The complexity of computing $\bar{\ell}_n(\lambda)$ is linear in the number of calibration samples, and the adjustment is constant, making $\hat{\alpha}(\lambda)$ itself efficient to evaluate.

Finally, we construct the miscoverage-regret frontier using the above estimator as follows: for each candidate robustness level $\lambda \in \Lambda$, we first solve the robust problem in (1) to obtain the decision rule $z_\lambda^*(\cdot)$, then evaluate the induced miscoverage and regret losses on calibration data. These empirical averages are corrected using the conservative adjustment in Proposition 3.3 to yield valid upper-bound estimates of risk. Collecting the pairs $(\hat{\alpha}_I(\lambda), \hat{\alpha}_R(\lambda))$ across all $\lambda$ forms the empirical frontier, which is then Pareto-pruned to remove dominated points.

We note that the primary computational bottleneck of the enumeration-based procedure is the need to solve (1) for each (discretized) element of the index set $\Lambda$. However, this cost is often manageable when $z_\lambda^*(\cdot)$ admits closed-form solutions or can be efficiently solved by off-the-shelf solvers, so this computational complexity is standard in the multi-objective optimization literature (Miettinen, 1999). We also validated this through our empirical analysis (Section 4.3).

### 3.2. Theoretical Analysis

We establish two key theoretical properties of the proposed estimator. First, we prove its validity, showing that under the exchangeability assumption, the proposed estimator $\hat{\alpha}_\ell(\lambda)$ upper bounds the true risk for any specification of the robustness parameter $\lambda$. Then, we demonstrate that under a majorant consistency assumption, the conservative trajectory traced by the estimator $(\hat{\alpha}_I(\lambda), \hat{\alpha}_R(\lambda))$ by varying $\lambda$ characterizes a certified Pareto frontier. All proofs in this section can be found in Appendix A through Appendix D.

**Theorem 3.4** (Validity). *Under Assumption 3.1 and 3.2, the estimator* $\hat{\alpha}_\ell(\lambda)$ *in* (5) *satisfies*

$$\mathbb{E}[\hat{\alpha}_\ell(\lambda)] \geq \mathbb{E}[\ell_\lambda(X_{n+1}, Y_{n+1})]. \tag{7}$$

Additionally, under the *i.i.d.* condition, we can establish a finite-sample error bound for the proposed estimator.

**Proposition 3.5** (Finite-sample error bound). *Under Assumption 3.2, suppose* $\{(x_i, y_i)\}_{i=1}^{n+1}$ *are i.i.d., then for any* $\delta > 0$ *and* $\lambda \in \Lambda$, *with probability at least* $1 - \delta$,

$$|\hat{\alpha}_\ell(\lambda) - \mathbb{E}[\ell_\lambda(X_{n+1}, Y_{n+1})]|$$
$$\leq \frac{B}{n+1}\left(\sqrt{\frac{n}{2}\log\left(\frac{2}{\delta}\right)} + 2 + \frac{1}{B}\right) =: \epsilon \tag{8}$$

Proposition 3.5 can be combined with Theorem 3.4 to produce a valid *one-shot* estimator with high probability by offsetting it with the error $\epsilon$ defined in (8):

$$\mathbb{P}\left\{\hat{\alpha}_\ell(\lambda) + \epsilon \geq \mathbb{E}\left[\ell(X_{n+1}, Y_{n+1})\right]\right\} \geq 1 - \delta.$$

Since this error bound decays to zero at an $O(n^{-1/2})$ parametric rate, this means that the estimator is sample efficient

and asymptotically consistent. We note that under pure exchangeability alone, a bound similar to Proposition 3.5 is generally not available. The issue is that exchangeability allows strong dependency, which could result in arbitrarily loose bounds.

Next, we show that the miscoverage-regret trade-off curve is (weakly) decreasing under the standard *majorant consistency* assumption (Delage & Ye, 2010; Rahimian & Mehrotra, 2019; Mohajerin Esfahani & Kuhn, 2018). This assumption guarantees that the trade-off curve preserves a monotone structure, thereby ensuring the existence of a well-defined Pareto frontier.

**Assumption 3.6** (Majorant consistency). For each robustness level $\lambda$ and context $x$, let $z_\lambda^*(x)$ be a measurable robust selector. For all $\lambda_1 \leq \lambda_2$, there is

$$\mathbb{E}\left[f\left(Y, z_{\lambda_1}^*(X)\right)\big|X\right] \leq \mathbb{E}\left[f\left(Y, z_{\lambda_2}^*(X)\right)\big|X\right] \text{ a.s.} \tag{9}$$

Assumption 3.6 describes the condition that shrinking the uncertainty set does not increase the conditional expected realized cost. It holds in a broad range of standard optimization problems, including robust linear optimization with polyhedral or ellipsoidal uncertainty sets (Ben-Tal & Nemirovski, 1999), robust quadratic optimization, regression with convex loss functions (*e.g.*, squared, absolute, or Huber loss) under norm-ball uncertainty (Ben-Tal & Nemirovski, 2001), and robust classification with convex surrogate losses (*e.g.*, hinge or logistic) (Bertsimas et al., 2011). It is also directly implied by DRO formulations with Wasserstein or $\phi$-divergence balls, where monotonicity of the robust objective in the size of the ball is well established (Delage & Ye, 2010). Nevertheless, we emphasize that this assumption is not required for the practical applicability of our algorithm. Even when the resulting trade-off curve is non-monotone and a Pareto frontier does not naturally arise, one can manually prune the constructed frontier to recover a well-defined Pareto frontier (Step 11 in Algorithm 1). For more intuition, we provide an example and a counterexample of Assumption 3.6 in Appendix C.

Combined with the properties of nested uncertainty sets and bounded regret in our problem setting, we can prove that the image of the trade-off curve coincides with the Pareto frontier of the robust policy family.

**Proposition 3.7** (True Pareto frontier). *Let* $\alpha_I(\lambda) := \mathbb{E}[I_\lambda(X, Y)]$ *and* $\alpha_R(\lambda) := \mathbb{E}[R_\lambda(X, Y)]$. *Under Assumption 3.6, for any* $\lambda_1 \leq \lambda_2$,

$$\alpha_I(\lambda_1) \geq \alpha_I(\lambda_2), \quad \alpha_R(\lambda_1) \leq \alpha_R(\lambda_2).$$

*Hence, the parametric curve* $\lambda \mapsto (\alpha_I(\lambda), \alpha_R(\lambda))$ *is (weakly) decreasing. Its image is the Pareto frontier within the policy family* $\{z_\lambda^* : \lambda \in \Lambda\}$.

**Algorithm 1** CREME

---

**Require:** Calibration data $\mathcal{D} := \{(x_i, y_i)\}_{i=1}^{n}$; robustness parameter set $\{\lambda_i\} \subseteq \Lambda$;

1: $\mathcal{I}_1, \mathcal{I}_2 \leftarrow$ Randomly split $\{1, \ldots, n\}$.
2: **for** $j \in \{1, 2\}$ **do**
3:     $\hat{\mathcal{F}}^{(j)} \leftarrow \emptyset$ initialize Pareto frontier;
4:     **for** $\lambda \in \{\lambda_i\}$ **do**
5:         $\bar{I}_n^{(j)}(\lambda) \leftarrow \sum_{i \in \mathcal{I}_j} I_\lambda(x_i, y_i)/|\mathcal{I}_j|$;
6:         $\bar{R}_n^{(j)}(\lambda) \leftarrow \sum_{i \in \mathcal{I}_j} R_\lambda(x_i, y_i)/|\mathcal{I}_j|$;
7:         $\tilde{\alpha}_R^{(j)}(\lambda) \leftarrow$ compute (6) by $\bar{R}_n^{(j)}(\lambda)$, $B = R_{\max}$;
8:         $\tilde{\alpha}_I^{(j)}(\lambda) \leftarrow$ compute (6) by $\bar{I}_n^{(j)}(\lambda)$, $B = 1$;
9:         $\hat{\mathcal{F}}^{(j)} \leftarrow \hat{\mathcal{F}}^{(j)} \cup \{(\hat{\alpha}_I^{(j)}(\lambda), \hat{\alpha}_R^{(j)}(\lambda))\}$;
10:     **end for**
11:     Remove dominated points from $\hat{\mathcal{F}}^{(j)}$;
12: **end for**
13: $\hat{\lambda} \leftarrow$ Decision maker selects $\lambda$ from observing $\hat{\mathcal{F}}^{(1)}$
14: **return** Post-hoc valid estimate $(\hat{\alpha}_I^{(2)}(\hat{\lambda}), \hat{\alpha}_R^{(2)}(\hat{\lambda}))$.

---

**Corollary 3.8** (Certified Pareto frontier). *Given that estimators* $\hat{\alpha}_I(\lambda), \hat{\alpha}_R(\lambda)$ *satisfy*

$$\hat{\alpha}_I(\lambda) \geq \alpha_I(\lambda), \quad \hat{\alpha}_R(\lambda) \geq \alpha_R(\lambda), \quad \forall \lambda \in \Lambda.$$

*Then the set* $\widehat{\mathcal{F}} := \left\{ (\hat{\alpha}_I(\lambda), \hat{\alpha}_R(\lambda)) : \lambda \in \Lambda \right\}$ *forms a conservative outer approximation of the true risk set.*

By Corollary 3.8, the lower-left envelope of $\widehat{\mathcal{F}}$ traces the certified Pareto frontier, thus providing guaranteed trade-offs between regret and miscoverage.

### 3.3. Re-calibration After Selection

In the previous sections, our analysis was carried out under the assumption that the robustness parameter $\lambda \in \Lambda$ is specified *a priori*. Under this setting, our estimator is used purely for *risk assessment*: for each candidate robustness level $\lambda$, it provides conservative estimates of the miscoverage and regret that would be incurred by deploying the corresponding robust decision. Our theoretical results establish that, when $\lambda$ is fixed in advance, the estimator enjoys exact finite-sample validity guarantees.

In practical decision-making settings, however, it is often desirable for the robustness parameter to be chosen *a posteriori*, after inspecting the estimated trade-offs. For example, after observing the estimated Pareto frontier $\hat{\mathcal{F}}$, a decision-maker may select a robustness level that optimizes a subjective miscoverage–regret preference,

$$\hat{\lambda} = \arg\max_{\lambda \in \Lambda} g\left(\hat{\alpha}_I(\lambda), \hat{\alpha}_R(\lambda)\right), \tag{10}$$

where $g : \mathbb{R}^2 \to \mathbb{R}$ encodes the decision-maker's risk preference. The selected robustness level $\hat{\lambda}$ is then deployed, and the corresponding estimated risks are reported.

This post-hoc selection step fundamentally alters the statistical setting. Because $\hat{\lambda}$ is chosen based on the same calibration data used to estimate risk, the loss sequence $\{\ell_{\hat{\lambda}}(X_i, Y_i)\}_{i=1}^{n+1}$ is no longer exchangeable. As a result, the exact validity guarantee of Theorem 3.4 may fail to hold. The following corollary formalizes this loss of validity.

**Corollary 3.9** (Post-hoc validity degradation). *Under Assumption 3.1 and Assumption 3.2, let* $\hat{\lambda}$ *be defined as in* (10). *Then the expectation of the estimator* $\hat{\alpha}_\ell(\hat{\lambda})$ *defined in* (5) *is lower bounded by*

$$\mathbb{E}\left[\hat{\alpha}_\ell(\hat{\lambda})\right] \geq \mathbb{E}\left[\ell_{\hat{\lambda}}(X_{n+1}, Y_{n+1})\right] - \Delta(g, \mathcal{P}),$$

*where the error term is defined as*

$$\Delta(g, \mathcal{P}) := \sup_{1 \leq i \leq n} \left| \mathbb{E}\left[\ell_{\hat{\lambda}}(X_i, Y_i)\right] - \mathbb{E}\left[\ell_{\hat{\lambda}}(X_{n+1}, Y_{n+1})\right] \right|.$$

Intuitively, the error term $\Delta(g, \mathcal{P})$ depends on how much asymmetry that selecting $\hat{\lambda}$ by $g$ introduces to the distribution $\mathcal{P}$. It arises because the data-dependent choice of $\hat{\lambda}$ leaks information from the calibration sample into the loss evaluation, thereby breaking the symmetry between calibration points and the test point. This violation of exchangeability is well known to undermine conformal-style guarantees (Barber et al., 2023). Consequently, the estimator may no longer be conservative after post-hoc selection. A detailed proof is provided in Appendix E.

We address this issue using a simple and effective data-splitting strategy (Cox, 1975; Rasines & Young, 2023), which restores exact finite-sample validity under post-hoc selection. Specifically, we randomly partition the calibration dataset $\{(X_i, Y_i)\}_{i=1}^{n}$ into two disjoint subsets $\mathcal{D}_1$ and $\mathcal{D}_2$. Using $\mathcal{D}_1$, we construct a *pre-hoc estimator* $\hat{\alpha}_\ell^{(1)}(\lambda)$ and form the corresponding Pareto frontier $\hat{\mathcal{F}}^{(1)}$. Only this pre-hoc Pareto frontier is revealed to the decision-maker and used to select the robustness level $\hat{\lambda}$ via (10). Independently, using $\mathcal{D}_2$, we construct a *post-hoc estimator* $\hat{\alpha}_\ell^{(2)}(\lambda)$, which is then evaluated at the selected robustness level $\hat{\lambda}$ to produce the re-calibrated miscoverage and regret estimates $\left(\hat{\alpha}_I^{(2)}(\hat{\lambda}), \hat{\alpha}_R^{(2)}(\hat{\lambda})\right)$. The pseudocode in Algorithm 1 summarizes our entire proposed method, which we term as Conformal REgret Miscoverage Estimate (CREME).

## 4. Experiments

This section presents the numerical evaluation of CREME (Algorithm 1). We focus on four representative optimization paradigms for decision-making problems: ($i$) linear programming; ($ii$) the newsvendor problem; ($iii$) portfolio optimization; and ($iv$) the shortest path problem. These problems are chosen because they exemplify widely studied optimization settings, and their regret can be efficiently computed using convex solvers. Their supremum $B$ defined in

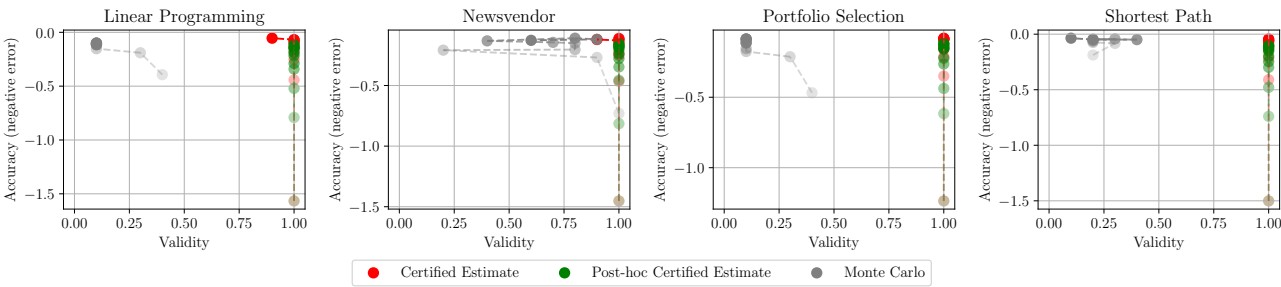

*Figure 3.* Validity-accuracy tradeoff curves under four optimization settings. Connected dots trace estimator performance as the number of calibration samples $n$ increases from 1 to 50, with greater opacity indicating smaller $n$. The proposed method is shown in red, and the baseline Monte Carlo estimator in gray. Both axes represent metrics where higher values indicate better performance ($\uparrow$), so methods appearing closer to the upper-right corner are more desirable.

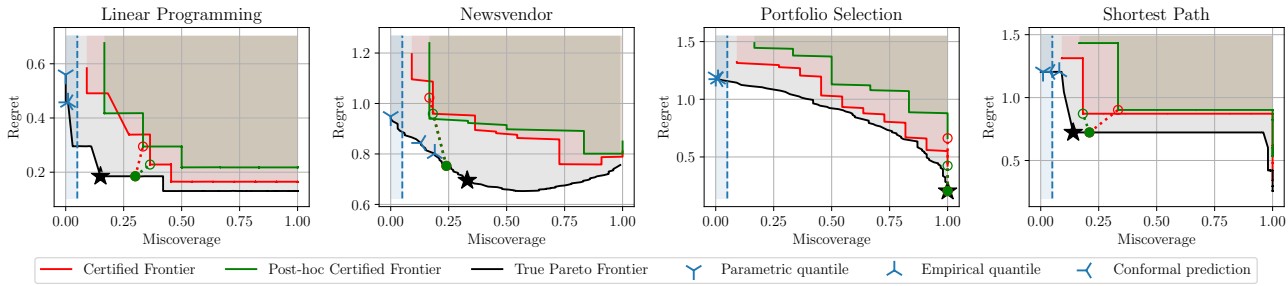

*Figure 4.* Miscoverage-regret tradeoff Pareto frontiers. Given some prespecified preference function, there is an optimal tradeoff point (black star) that can be computed from the true Pareto frontier (black line). We can obtain a certified tradeoff point (red dot) and a post-hoc tradeoff point (green dot) by using CREME. Blue markers represent tradeoff points derived from the three baseline methods that identify $\lambda$ values ensuring the miscoverage rate remains below 5% (vertical dashed blue line).

Assumption 3.2 is either known by design or approximated through the empirical maximum $\max_{1 \leq i \leq n} \ell_\lambda(x_i, y_i)$. Additional experimental configuration details, explanations, and additional experiment results can be found in Appendix G.

### 4.1. Validity-Accuracy Analysis

We first evaluate the quality of the miscoverage-regret frontier produced by CREME using two metrics: *validity* and *accuracy*. Validity measures whether the estimator successfully upper bounds the true risk, defined as the proportion of trials in which both $\hat{\mathbb{E}}[\hat{\alpha}_R(\lambda)]$ and $\hat{\mathbb{E}}[\hat{\alpha}_I(\lambda)]$ are upper bounding estimates [1]. Accuracy assesses the tightness of this bound, defined as the negative average distance between $(\hat{\alpha}_I(\lambda), \hat{\alpha}_R(\lambda))$ and $(\alpha_I(\lambda), \alpha_R(\lambda))$. They are computed from averaging across 20 independent trials.

Figure 3 presents the evaluation results comparing CREME with the empirical Monte Carlo estimator (*i.e.*, $\bar{\ell}_n(\lambda)$). It can be seen that across all optimization settings, the Monte Carlo estimator (gray) tends to lie toward the left of the plot, while CREME (red and green) concentrates in the upper-right region. This difference arises because naive baselines

are primarily designed for consistent estimation, achieving high accuracy without validity guarantees. In contrast, our proposed procedure strikes a balanced trade-off, maintaining strong performance on both validity and accuracy. These results highlight that CREME is well-suited for high-stakes decision-making tasks where both accuracy and validity are critical.

Figure 3 also provides insights into the finite-sample behavior of CREME. In the small-sample regime with a small $n$, CREME exhibits relatively low accuracy but high valid portion, positioning itself in the lower-right region of the plot. This behavior is desirable: under uncertainty and limited resources, CREME prioritizes conservatism by erring on the side of validity. As $n$ increases, the trajectory rises at around an exponential rate along the accuracy axis while maintaining full validity. The trajectory's accuracy eventually converges to 0, demonstrating asymptotic consistency.

### 4.2. Decision Quality Evaluation

We also assess the quality of the robustness parameter selected based on CREME. We show that given any prespecified weight that encodes the decision-maker's preference for the miscoverage-regret tradeoff, CREME can identify $\hat{\lambda}$ that closely approximates the optimal $\lambda^*$, providing

---

[1] $\hat{\mathbb{E}}[\cdot]$ is computed as the sample average over 100 iterations.

*Table 1.* Sensitivity analysis of `CREME` against number of samples $n$ and the robustness level index set size $|\Lambda|$.

| | | Linear Programming | | Newsvendor | | Portfolio Optimization | | Shortest Path | |
|---|---|---|---|---|---|---|---|---|---|
| | | Gap | Time (s) | Gap | Time (s) | Gap | Time (s) | Gap | Time (s) |
| | $n = 10$ | $0.10 \pm 0.03$ | $0.58 \pm 0.03$ | $0.19 \pm 0.08$ | $0.00 \pm 0.00$ | $0.16 \pm 0.07$ | $0.84 \pm 0.10$ | $0.11 \pm 0.03$ | $0.72 \pm 0.04$ |
| $|\Lambda| = 10$ | $n = 20$ | $0.09 \pm 0.03$ | $0.81 \pm 0.03$ | $0.12 \pm 0.04$ | $0.00 \pm 0.00$ | $0.13 \pm 0.07$ | $1.14 \pm 0.12$ | $0.06 \pm 0.01$ | $0.99 \pm 0.06$ |
| | $n = 30$ | $0.08 \pm 0.01$ | $1.00 \pm 0.04$ | $0.13 \pm 0.04$ | $0.00 \pm 0.00$ | $0.09 \pm 0.02$ | $1.50 \pm 0.13$ | $0.06 \pm 0.01$ | $1.20 \pm 0.07$ |
| | $n = 10$ | $0.10 \pm 0.03$ | $0.58 \pm 0.03$ | $0.19 \pm 0.08$ | $0.00 \pm 0.00$ | $0.16 \pm 0.07$ | $0.84 \pm 0.10$ | $0.11 \pm 0.03$ | $0.72 \pm 0.04$ |
| $|\Lambda| = 20$ | $n = 20$ | $0.09 \pm 0.03$ | $0.81 \pm 0.03$ | $0.12 \pm 0.04$ | $0.00 \pm 0.00$ | $0.13 \pm 0.07$ | $1.14 \pm 0.12$ | $0.06 \pm 0.01$ | $0.99 \pm 0.06$ |
| | $n = 30$ | $0.08 \pm 0.01$ | $1.00 \pm 0.04$ | $0.13 \pm 0.04$ | $0.00 \pm 0.00$ | $0.09 \pm 0.02$ | $1.50 \pm 0.13$ | $0.06 \pm 0.01$ | $1.20 \pm 0.07$ |
| | $n = 10$ | $0.09 \pm 0.03$ | $1.73 \pm 0.06$ | $0.16 \pm 0.07$ | $0.00 \pm 0.00$ | $0.17 \pm 0.08$ | $2.50 \pm 0.28$ | $0.11 \pm 0.04$ | $2.13 \pm 0.08$ |
| $|\Lambda| = 30$ | $n = 20$ | $0.08 \pm 0.02$ | $2.42 \pm 0.09$ | $0.11 \pm 0.04$ | $0.00 \pm 0.00$ | $0.13 \pm 0.07$ | $3.43 \pm 0.34$ | $0.05 \pm 0.01$ | $2.93 \pm 0.15$ |
| | $n = 30$ | $0.06 \pm 0.02$ | $2.98 \pm 0.13$ | $0.11 \pm 0.04$ | $0.00 \pm 0.00$ | $0.08 \pm 0.03$ | $4.49 \pm 0.44$ | $0.04 \pm 0.01$ | $3.63 \pm 0.25$ |

high-quality robustness calibration for decision-making.

We include three baselines: ($i$) Parametric quantile: chooses $\hat{\lambda}$ as the 95%-quantile of $\|Y - \mu\|_\infty$ assuming $Y \sim \mathcal{N}(\hat{\mu}, \hat{\Sigma})$, with $\hat{\mu}$ and $\hat{\Sigma}$ estimated from the data. ($ii$) Empirical quantile: $\lambda$ is taken as the 95% empirical quantile of $|y_i - \bar{y}|$. ($iii$) Conformal prediction: uses $\mathbb{E}[Y]$ as the base predictor and construct 95%-valid $\ell_\infty$-ball conformal prediction set, where $\hat{\lambda}$ is defined as its radius (Patel et al., 2024b). These baselines represent the typical practice for robustness parameter selection, where the user would first specify a small miscoverage level (*e.g.*, 5%) and then determine a data-driven $\hat{\lambda}$ that achieves the desired coverage. The ground truth Pareto frontier is approximated via 100 samples from the true distribution enumerating over $\lambda \in \Lambda$.

Figure 4 visualizes the evaluated Pareto frontier and tradeoff outcome. It can be seen that the estimated Pareto frontiers (red and green) lie closely above the true frontier (black), with the post-hoc certified frontier (green) consistently lying slightly higher above its standard counterpart, providing a slightly conservative but safer miscoverage-regret tradeoff characterization. Both select robustness level that is close to $\lambda^*$ (black dot) and showcase similar miscoverage-regret tradeoff. On the other hand, the tradeoff points selected by the three baselines achieve low miscoverage but suffer from high regret. This is because these baselines, similar to many existing robust set specifying heuristics, are designed to conservatively control miscoverage and thus myopically select suboptimal robustness parameters, leading to an undesirable miscoverage-regret tradeoff selection. On the other hand, `CREME` is capable of guiding the selection of appropriate robustness parameters that closely resemble its true optimal value, supporting a balanced decision calibration.

On a side note, observe in Figure 4 that the ground-truth curve for the newsvendor and portfolio selection problems contains points that are not strictly on the Pareto frontier (*i.e.*, non-dominant points). This observation is consistent with Assumption 3.6 and Proposition 3.7, so additional pruning must be done if one desires the full Pareto frontier.

### 4.3. Ablation Studies

Finally, we conduct an ablation study on `CREME`'s performances under different hyperparameter specifications: the number of samples $n$ and the robustness level index set size $|\Lambda|$. We examine two performance metrics: ($i$) Gap: the approximation gap of the estimated Pareto frontier to its true counterpart, measured by their average distance (*i.e.*, negative of the accuracy metric); ($ii$) Time (s): the execution time (seconds) for `CREME` to construct a full estimated Pareto frontier. Numbers are reported as mean $\pm$ standard deviation computed from 20 trials.

Table 1 summarizes the experimental results. As $n$ and $|\Lambda|$ increase, the approximation gap decreases, consistent with our theoretical bound in Proposition 3.5. At the same time, larger $n$ and $|\Lambda|$ lead to increased runtime, which peaks at approximately 5 seconds across all configurations. This computational cost is typically manageable in practice since the construction of the frontier is usually a one-shot procedure. In particular, the runtime for the newsvendor setting remains uniformly low, owing to the availability of closed-form solutions. Overall, these results demonstrate that our approach achieves both strong empirical performance and computational efficiency suitable for practical deployment.

## 5. Conclusion and Discussions

We proposed `CREME`, a conformalized framework for robust decision-making that certifies both miscoverage and regret across robustness levels, enabling practitioners to select appropriate robustness in optimization design. Theoretically, we showed that under mild conditions, the estimator provides conservative, distribution-free finite-sample guarantees and traces a certified Pareto frontier for the miscoverage–regret tradeoff. Across four optimization implementations, `CREME` produces frontiers with high validity and accuracy and guides robustness selection to balance miscoverage and regret. Thus, `CREME` offers a principled alternative to ad hoc calibration, addressing a key gap in the existing literature.

There are two limitations with this work: ($i$) Our theoretical

analysis relies on knowing the upper bound $B$ of the loss function (Assumption 3.2), which inherits from (Angelopoulos et al., 2022). However, we noticed that in practice $B$ can be replaced with data-driven estimates without significant harm to the performance (see Appendix G), which motivates potential future work to principally analyze these estimators and develop new conformal algorithms that avoid the bounded loss assumption altogether. ($ii$) Our work uses data splitting to address post-hoc invalidity, which comes at the cost of reducing sample efficiency. We refer readers to (Berk et al., 2013; Zrnic & Jordan, 2023) for recent developments in post-hoc inference, and expect future work to integrate these techniques to further improve sample efficiency[2].

Additionally, there are two other promising extensions of the current framework, including ($i$) extending the framework to nonexchangeable settings (*e.g.*, time series), and ($iii$) maintaining validity in the presence of optimization solver error. For ($i$), we hypothesize that a weighted version of our estimator, using weights similar to those in (Tibshirani et al., 2019), could be developed to account for non-exchangeability in the data; To achieve ($iii$), one can introduce an offset term that scales with the solver suboptimality gap (see Appendix F). We leave the details to be developed in potential future works.

## Acknowledgements

The authors acknowledge support from the 2024 Block Center Seed Fund at Carnegie Mellon University, which partially supported this research.

## Impact Statement

This paper presents work whose goal is to advance the field of machine learning. There are many potential societal consequences of our work, none of which we feel must be specifically highlighted here.

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

## A. Proof of Theorem 3.4

*Proof.* We prove by showing:

$$\mathbb{E}\left[\hat{\alpha}_\ell(\lambda)\right] \overset{(a)}{\geq} \mathbb{E}\left[\tilde{\alpha}_\ell(\lambda)\right] \overset{(b)}{\geq} \mathbb{E}[\ell_\lambda(X_{n+1}, Y_{n+1})]. \tag{11}$$

To prove $(a)$, notice that $\tilde{\alpha}_\ell(\lambda)$ is equivalent to (5) after removing the ceiling operator, and by

$$\frac{(n+1)(B - \hat{\alpha}_\ell(\lambda))}{n} \leq \frac{\lceil (n+1)(B - \hat{\alpha}_\ell(\lambda)) \rceil}{n} \overset{(5)}{\leq} B - \bar{\ell}_n(\lambda),$$

therefore $\hat{\alpha}_\ell(\lambda) \geq \tilde{\alpha}_\ell(\lambda)$ almost surely, which implies $(a)$. To prove $(b)$, notice that Assumption 3.1 implies

$$\mathbb{E}\left[\ell_\lambda(X_1, Y_1)\right] = \ldots = \mathbb{E}\left[\ell_\lambda(X_n, Y_n)\right] = \mathbb{E}\left[\ell_\lambda(X_{n+1}, Y_{n+1})\right], \tag{12}$$

we then expand the difference of two sides of $(b)$ by using the definition of $\tilde{\alpha}_\ell(\lambda)$ in (6):

$$\mathbb{E}\left[\tilde{\alpha}_\ell(\lambda)\right] - \mathbb{E}[\ell_\lambda(X_{n+1}, Y_{n+1})] = \frac{1}{n+1}\sum_{i=1}^{n}\mathbb{E}[\ell_\lambda(X_i, Y_i)] + \frac{B}{n+1} - \mathbb{E}[\ell_\lambda(X_{n+1}, Y_{n+1})]$$

$$= \frac{n}{n+1}\mathbb{E}[\ell_\lambda(X_{n+1}, Y_{n+1})] + \frac{B}{n+1} - \mathbb{E}[\ell_\lambda(X_{n+1}, Y_{n+1})] = \frac{B - \mathbb{E}[\ell_\lambda(X_{n+1}, Y_{n+1})]}{n+1} \geq 0.$$

where the last inequality follows from Assumption 3.2. This proves $(b)$ holds and thus concludes the proof. □

## B. Proof of Proposition 3.5

*Proof of Proposition 3.5.* Notice the following decomposition:

$$|\hat{\alpha}_\ell(\lambda) - \mathbb{E}[\ell_\lambda(X_{n+1}, Y_{n+1})]| \leq |\hat{\alpha}_\ell(\lambda) - \tilde{\alpha}_\ell(\lambda)| + |\tilde{\alpha}_\ell(\lambda) - \mathbb{E}[\ell_\lambda(X_{n+1}, Y_{n+1})]|.$$

The first term can be upper-bounded by $1/(n+1)$ by the definition of $\hat{\alpha}_\ell(\lambda)$ and $\tilde{\alpha}_\ell(\lambda)$. The second term can be further expanded as

$$|\tilde{\alpha}_\ell(\lambda) - \mathbb{E}[\ell_\lambda(X_{n+1}, Y_{n+1})]| = \left| \frac{n}{n+1}\bar{\ell}_n(\lambda) + \frac{B}{n+1} - \mathbb{E}[\ell_\lambda(X_{n+1}, Y_{n+1})] \right|$$

$$= \left| \frac{n}{n+1}\left\{ \bar{\ell}_n(\lambda) - \mathbb{E}[\ell_\lambda(X_{n+1}, Y_{n+1})] \right\} + \frac{B}{n+1} - \frac{1}{n+1}\mathbb{E}[\ell_\lambda(X_{n+1}, Y_{n+1})] \right|$$

$$\leq \frac{n}{n+1}\left| \bar{\ell}_n(\lambda) - \mathbb{E}[\ell_\lambda(X_{n+1}, Y_{n+1})] \right| + \frac{2B}{n+1},$$

where for the last inequality we used the bounding condition (Assumption 3.2). Observe that by the *i.i.d.* condition, using Hoeffding's inequality, we know that for any $\delta > 0$, there is

$$\left| \bar{\ell}_n(\lambda) - \mathbb{E}[\ell_\lambda(X_{n+1}, Y_{n+1})] \right| \leq B \cdot \sqrt{\frac{1}{2n}\log\left(\frac{2}{\delta}\right)}, \quad w.p. \geq 1 - \delta.$$

Plugging this error term into the expansion above, we get

$$|\tilde{\alpha}_\ell(\lambda) - \mathbb{E}[\ell_\lambda(X_{n+1}, Y_{n+1})]| \leq \frac{nB}{n+1}\sqrt{\frac{1}{2n}\log\left(\frac{2}{\delta}\right)} + \frac{2B}{n+1} = \frac{B}{n+1}\left( \sqrt{\frac{n}{2}\log\left(\frac{2}{\delta}\right)} + 2 \right), \quad w.p. \geq 1 - \delta.$$

Therefore, plugging back into the original equation, we know that with probability no less that $1 - \delta$,

$$|\hat{\alpha}_\ell(\lambda) - \mathbb{E}[\ell_\lambda(X_{n+1}, Y_{n+1})]| \leq \frac{B}{n+1}\left( \sqrt{\frac{n}{2}\log\left(\frac{2}{\delta}\right)} + 2 \right) + \frac{1}{n+1} = \frac{B}{n+1}\left( \sqrt{\frac{n}{2}\log\left(\frac{2}{\delta}\right)} + 2 + \frac{1}{B} \right).$$

We've concluded the proof. □

Using Proposition 3.5, we can further establish a reliable lower bound on the difference in the true expected loss function between two robustness parameters, as summarized below.

**Corollary B.1.** *Given $\lambda_1, \lambda_2 \in \Lambda$ and $\lambda_1 \neq \lambda_2$, then with probability no less than $1 - \delta$, there is*

$$|\mathbb{E}[\ell_{\lambda_1}(X_{n+1}, Y_{n+1})] - \mathbb{E}[\ell_{\lambda_2}(X_{n+1}, Y_{n+1})]| \geq |\hat{\alpha}_\ell(\lambda_1) - \hat{\alpha}_\ell(\lambda_2)| - \frac{2B}{n+1}\left(\sqrt{\frac{n}{2}\log\left(\frac{1}{\delta}\right)} + 2 + \frac{1}{B}\right).$$

*Proof of Corollary B.1.* Since for any $\lambda \in \Lambda$, we know from Proposition 3.5 that with probability $1 - \delta'$, there is

$$|\mathbb{E}[\ell_\lambda(X_{n+1}, Y_{n+1})] - \hat{\alpha}_\ell(\lambda)| \leq \frac{B}{n+1}\left(\sqrt{\frac{n}{2}\log\left(\frac{2}{\delta'}\right)} + 2 + \frac{1}{B}\right).$$

Therefore, by setting $\delta' = \delta/2$, we know that with probability at least $1 - \delta$, there is

$$
\begin{aligned}
&|\mathbb{E}[\ell_{\lambda_1}(X_{n+1}, Y_{n+1})] - \mathbb{E}[\ell_{\lambda_2}(X_{n+1}, Y_{n+1})] - (\hat{\alpha}_\ell(\lambda_1) - \hat{\alpha}_\ell(\lambda_2))| \\
&\leq |\mathbb{E}[\ell_{\lambda_1}(X_{n+1}, Y_{n+1})] - \hat{\alpha}_\ell(\lambda_1)| + |\mathbb{E}[\ell_{\lambda_2}(X_{n+1}, Y_{n+1})] - \hat{\alpha}_\ell(\lambda_2)| \\
&\leq \frac{2B}{n+1}\left(\sqrt{\frac{n}{2}\log\left(\frac{1}{\delta}\right)} + 2 + \frac{1}{B}\right).
\end{aligned}
$$

Rearranging, we get:

$$|\mathbb{E}[\ell_{\lambda_1}(X_{n+1}, Y_{n+1})] - \mathbb{E}[\ell_{\lambda_2}(X_{n+1}, Y_{n+1})]| \geq |\hat{\alpha}_\ell(\lambda_1) - \hat{\alpha}_\ell(\lambda_2)| - \frac{2B}{n+1}\left(\sqrt{\frac{n}{2}\log\left(\frac{1}{\delta}\right)} + 2 + \frac{1}{B}\right).$$

This finishes the proof. $\square$

The difference in the true expected loss function between two robustness parameters is particularly useful in practice, as it reflects how much gain or loss a decision-maker can expect when switching from $\lambda_1$ to $\lambda_2$, as illustrated in Figure 1. Corollary B.1 suggests that we can simply use the difference in estimators after offsetting with the error term $2\epsilon$ to obtain a lower bound, which informs the decision maker of a conservative estimate of the change magnitude. Practically, this can help reliably assess the trade-offs involved in selecting different robustness levels, leading to more principled and robust decision-making.

## C. Discussion of Assumption 3.6

To better understand the scope and limitations of our assumptions, we provide two contrasting examples. The first demonstrates a setting in which all conditions are satisfied, showing that the proposition applies naturally in classical robust optimization problems. The second example highlights a case where the key *majorant consistency* assumption fails, which happens when actuation penalties are asymmetric or the predictive model is misspecified, illustrating the boundaries of our results and where caution is required in practice. Without loss of generality, we assume $X$ to be degenerate in both examples, so that all arguments are marginalized with respect to $Y$.

**Example (assumption holds)** Consider a linear optimization with a compact convex feasible region

$$\min_z \langle Y, z \rangle \quad \text{s.t.} \quad \|z\|_2 \leq M,$$

and assume $\|Y\| \leq K$ a.s. (hence regret is uniformly bounded) and that measurable selectors are taken. We define nested uncertainty sets $\mathcal{U}_\lambda = \{y : \|y - \mu\|_2 \leq \lambda\}$, which by definition guarantees that the miscoverage is a decreasing function of $\lambda$. We define the robust variant of the optimization problem as

$$\min_z \max_{y \in \mathcal{U}_\lambda} \langle y, z \rangle \quad \text{s.t.} \quad \|z\|_2 \leq M.$$

Note that the inner maximization simplifies to the support function of the ball, *i.e.*, $\langle \mu, z \rangle + \lambda \|z\|_2$, so plugging this result in and solving for the outer minimization problem gets

$$
z_\lambda^* = \begin{cases} 0, & \lambda \geq \|\mu\|_2, \\ -M \cdot \mu/\|\mu\|_2, & \lambda < \|\mu\|_2. \end{cases}
$$

Therefore, $\mathbb{E}[f(Y, z_\lambda^*)] = \langle \mu, z_\lambda^* \rangle \in \{0, -M\|\mu\|_2\}$ is weakly increasing as $\lambda$ increasing, satisfying Assumption 3.6. Nesting holds by construction, and

$$
R_\lambda(X, Y) = \langle Y, z_\lambda^* \rangle - \min_{\|z\| \leq M} \langle Y, z \rangle \leq 2M\|Y\| \leq 2MK,
$$

so $R_\lambda(X, Y) \leq B$ by setting $B = 2MK$, and it is easy to verify that $R_\lambda(X, Y) \leq B$ is an increasing function of $\lambda$, therefore the statements in Proposition 3.7 hold. In fact, the frontier in this example is *strictly decreasing* whenever $\mathbb{P}\{\|\mu\|_2 > \lambda\} > 0$ for some $\lambda$. This example illustrates that the assumption is mild and naturally satisfied in common robust linear programs with ellipsoidal uncertainty and bounded outcomes.

*Remark* C.1. Note that this example naturally extends to the conditional setting, where $\mu$ is replaced by some estimator $\hat{g}(x)$ based on the covariate random variable $X$, and $\mathbb{E}[f(Y, z_\lambda^*)]$ is replaced by $\mathbb{E}[f(Y, z_\lambda^*) \mid X]$. All the arguments above remain true.

**Counterexample (assumption fails)** Let $Y \in \{-1, +1\}$ with $\mathbb{P}\{Y = +1\} = 0.9$, and decision variable space $\mathcal{Z} = [-1, 1]$. Define the objective function as

$$
f(y, z) = \begin{cases} (z - y)^2 & \text{if } z \in [0, 1], \\ (z - y)^2 + \frac{1}{4} & \text{if } z \in [-1, 0). \end{cases}
$$

Define $\Lambda = \{\lambda_1, \lambda_2\}$ with $\lambda_1 < \lambda_2$, and nested uncertainty sets $\mathcal{U}_{\lambda_1} = \{-1\}$ and $\mathcal{U}_{\lambda_2} = \{-1, +1\}$. It can be seen that the (robust) solutions to $\min_z \max_{y \in \mathcal{U}_\lambda} f(y, z)$ can be divided into two cases: (*i*) When $\lambda = \lambda_1$, $z_{\lambda_1}^* = -1$; (*ii*) When $\lambda = \lambda_2$, $z_{\lambda_2}^* = 0$. Therefore, it can be seen that

$$
\mathbb{E}[f(Y, z_{\lambda_1}^*)] = 0.9 \cdot (4 + 0.25) + 0.1 \cdot 0.25 = \mathbf{3.85} > \mathbb{E}[f(Y, z_{\lambda_2}^*)] = 0.9 \cdot 1 + 0.1 \cdot 1 = \mathbf{1},
$$

which makes this optimization problem violate Assumption 3.6.

In the meantime, in this counterexample, while miscoverage is still a decreasing function of $\lambda$ (from 0.9 under $\lambda_1$ to 0 under $\lambda_2$), the expected regret is no longer an increasing function of $\lambda$. Specifically, since $\min_{z \in [-1,1]} f(+1, z) = 0$ (achieved at $z = 1$) and $\min_{z \in [-1,1]} f(-1, z) = 0.25$ (achieved at $z = -1$), there is

$$
\alpha_R(\lambda_1) = \mathbb{E}\left[(4 + 0.25 - 0)\,\mathbb{1}\{Y = +1\} + (0.25 - 0.25)\,\mathbb{1}\{Y = -1\}\right] = 3.6 + 0.9 \cdot 0.25,
$$
$$
\alpha_R(\lambda_2) = \mathbb{E}\left[(1 - 0)\,\mathbb{1}\{Y = +1\} + (1 - 0.25)\,\mathbb{1}\{Y = -1\}\right] = 0.9 + 0.1 \cdot (1 - 0.25),
$$

and hence $\alpha_R(\lambda_1) > \alpha_R(\lambda_2)$, making expected regret an decreasing function of $\lambda$.

In sum, in this example, all other conditions (nesting, measurability, bounded regret) hold, but the proposition fails without majorant consistency. This illustrates that asymmetric model/actuation costs or misspecified robust surrogates can break the decreasing trade-off.

# D. Proof of Certified Pareto Frontier

*Proof of Proposition 3.7.* By nesting, for any $(X, Y)$ and $\lambda_1 \leq \lambda_2$, $\mathcal{U}_{\lambda_1}(X) \subseteq \mathcal{U}_{\lambda_2}(X)$, hence

$$
\mathbb{1}\{Y \notin \mathcal{U}_{\lambda_1}(X)\} \geq \mathbb{1}\{Y \notin \mathcal{U}_{\lambda_2}(X)\}.
$$

Taking expectations on both sides yields $\alpha_I(\lambda_1) \geq \alpha_I(\lambda_2)$.

Observe that

$$
\alpha_R(\lambda) = \mathbb{E}\left[f\left(Y, z_\lambda^*(X)\right)\right] - \mathbb{E}\left[\min_{z \in \mathcal{Z}} f(Y, z)\right].
$$

The second term is independent of $\lambda$. By the law of iterated expectations and Assumption 3.6,

$$\mathbb{E}\left[f\left(Y, z_{\lambda_2}^*(X)\right)\right] = \mathbb{E}\left[\mathbb{E}\left[f\left(Y, z_{\lambda_2}^*(X)\right)\big|X\right]\right] \geq \mathbb{E}\left[\mathbb{E}\left[f\left(Y, z_{\lambda_1}^*(X)\right)\big|X\right]\right] = \mathbb{E}\left[f\left(Y, z_{\lambda_1}^*(X)\right)\right].$$

Therefore $\alpha_R(\lambda_1) \leq \alpha_R(\lambda_2)$.

Because as $\lambda$ increases (*i.e.*, $\mathcal{U}_\lambda$ expands), $\alpha_I$ weakly decreases while $\alpha_R$ weakly increases, the map $\lambda \mapsto (\alpha_I(\lambda), \alpha_R(\lambda))$ is (weakly) decreasing. Fix $\lambda \in \Lambda$. For any $\lambda' \neq \lambda$, if $\lambda' < \lambda$ then $\alpha_I(\lambda') \geq \alpha_I(\lambda)$ and $\alpha_R(\lambda') \leq \alpha_R(\lambda)$; if $\lambda' > \lambda$ the inequalities reverse. Hence no policy in the family $\{z_\lambda^*\}$ strictly improves both coordinates over $z_\lambda^*$. Therefore every image point is Pareto efficient, and the image is exactly the Pareto frontier achievable by this policy family. (Ties may occur if the inequalities become equalities for some $\lambda_1 < \lambda_2$.) $\qquad\square$

*Proof of Corollary 3.8.* For a given $\lambda$, the coordinatewise inequalities imply $(\alpha_I(\lambda), \alpha_R(\lambda)) \preceq (\hat{\alpha}_I(\lambda), \hat{\alpha}_R(\lambda))$ in the product order on $\mathbb{R}^2$. Hence $\widehat{\mathcal{S}}$ is an outer (safe) approximation of the true risk set $\mathcal{S} := \{(\alpha_I(\lambda), \alpha_R(\lambda)) : \lambda \in \Lambda\}$. Taking the set of non-dominated points (lower-left envelope) of $\widehat{\mathcal{S}}$ yields certified trade-offs: any selected $\lambda$ on this envelope admits guarantees $\mathbb{E}[I_\lambda(X, Y)] \leq \hat{\alpha}_I(\lambda)$ and $\mathbb{E}[R_\lambda(X, Y)] \leq \hat{\alpha}_R(\lambda)$ (by the direction of the bounds), *i.e.*, a certified Pareto frontier. $\qquad\square$

# E. Proof of Corollary 3.9

*Proof of Corollary 3.9.* By how $\hat{\lambda}$ is constructed, it is a random variable measurable with respect to the sigma field generated by $\{(X_i, Y_i)\}_{i=1}^n$. Therefore, $\{\ell_{\hat{\lambda}}(X_i, Y_i)\}_{i=1}^{n+1} \mid \hat{\lambda}$ is not an exchangeable sequence, which breaks (12). By using the definition of $\tilde{\alpha}_\ell(\lambda)$ in (6), there is:

$$\begin{aligned}
&\mathbb{E}\left[\tilde{\alpha}_\ell(\hat{\lambda})\right] - \mathbb{E}[\ell_{\hat{\lambda}}(X_{n+1}, Y_{n+1})] \\
&= \frac{1}{n+1}\sum_{i=1}^n \mathbb{E}[\ell_{\hat{\lambda}}(X_i, Y_i)] + \frac{B}{n+1} - \mathbb{E}[\ell_{\hat{\lambda}}(X_{n+1}, Y_{n+1})] \\
&= \frac{1}{n+1}\sum_{i=1}^n \mathbb{E}[\ell_{\hat{\lambda}}(X_i, Y_i)] + \frac{B}{n+1} - \frac{1}{n+1}\sum_{i=1}^{n+1} \mathbb{E}[\ell_{\hat{\lambda}}(X_{n+1}, Y_{n+1})] \\
&= \frac{B - \mathbb{E}[\ell_{\hat{\lambda}}(X_{n+1}, Y_{n+1})]}{n+1} + \frac{1}{n+1}\sum_{i=1}^n \left\{\mathbb{E}[\ell_{\hat{\lambda}}(X_i, Y_i)] - \mathbb{E}[\ell_{\hat{\lambda}}(X_{n+1}, Y_{n+1})]\right\} \\
&\geq -\frac{1}{n+1}\sum_{i=1}^n \left|\mathbb{E}[\ell_{\hat{\lambda}}(X_i, Y_i)] - \mathbb{E}[\ell_{\hat{\lambda}}(X_{n+1}, Y_{n+1})]\right| \\
&\geq -\frac{n}{n+1}\sup_{1 \leq i \leq n} \left|\mathbb{E}[\ell_{\hat{\lambda}}(X_i, Y_i)] - \mathbb{E}[\ell_{\hat{\lambda}}(X_{n+1}, Y_{n+1})]\right| \\
&\geq -\sup_{1 \leq i \leq n} \left|\mathbb{E}[\ell_{\hat{\lambda}}(X_i, Y_i)] - \mathbb{E}[\ell_{\hat{\lambda}}(X_{n+1}, Y_{n+1})]\right|.
\end{aligned}$$

The first inequality follows from Assumption 3.2. Therefore, using the chain equality (11), we conclude that

$$\mathbb{E}\left[\hat{\alpha}_\ell(\lambda)\right] \geq \mathbb{E}[\ell_{\hat{\lambda}}(X_{n+1}, Y_{n+1})] - \sup_{1 \leq i \leq n} \left|\mathbb{E}[\ell_{\hat{\lambda}}(X_i, Y_i)] - \mathbb{E}[\ell_{\hat{\lambda}}(X_{n+1}, Y_{n+1})]\right|.$$

This finishes the proof. $\qquad\square$

*Remark* E.1. The splitting procedure (Algorithm 1) can retain the exact validity guarantee as in Theorem 3.4 because conditional exchangeability is preserved, and therefore (12) holds. Specifically, given two random splits from the calibration data, denoted by the index sets $\mathcal{I}_1$ and $\mathcal{I}_2$, then by construction of $\hat{\alpha}_\ell^{(1)}(\lambda)$ and $\hat{\mathcal{F}}^{(1)}$, $\hat{\lambda}$ is a random variable that is measurable with respect to the sigma field generated by $\{(X_i, Y_i)\}_{i \in \mathcal{I}_1}$. Therefore, there is

$$\left\{\ell_{\hat{\lambda}}(X_i, Y_i)\right\}_{i \in \mathcal{I}_2} \mid \{(X_i, Y_i)\}_{i \in \mathcal{I}_1} \quad \text{is exchangeable} \implies \left\{\ell_{\hat{\lambda}}(X_i, Y_i)\right\}_{i \in \mathcal{I}_2} \mid \hat{\lambda} \quad \text{is exchangeable.}$$

This implies the following equality chain

$$\mathbb{E}\left[\ell_{\hat{\lambda}}(X_{i_1}, Y_{i_1}) \mid \hat{\lambda}\right] = \ldots = \mathbb{E}\left[\ell_{\hat{\lambda}}(X_{i_{|\mathcal{I}_2|}}, Y_{i_{|\mathcal{I}_2|}}) \mid \hat{\lambda}\right],$$

Therefore, conditioning on any $\hat{\lambda}$, there is

$$\mathbb{E}[\tilde{\alpha}_\ell(\hat{\lambda}) \mid \hat{\lambda}] - \mathbb{E}[\ell_{\hat{\lambda}}(X_{n+1}, Y_{n+1}) \mid \hat{\lambda}] = \frac{1}{|\mathcal{I}_2| + 1} \sum_{i \in \mathcal{I}_2} \mathbb{E}[\ell_{\hat{\lambda}}(X_i, Y_i) \mid \hat{\lambda}] + \frac{B}{|\mathcal{I}_2| + 1} - \mathbb{E}[\ell_{\hat{\lambda}}(X_{n+1}, Y_{n+1}) \mid \hat{\lambda}]$$

$$= \frac{|\mathcal{I}_2|}{|\mathcal{I}_2| + 1} \mathbb{E}[\ell_{\hat{\lambda}}(X_{n+1}, Y_{n+1}) \mid \hat{\lambda}] + \frac{B}{|\mathcal{I}_2| + 1} - \mathbb{E}[\ell_{\hat{\lambda}}(X_{n+1}, Y_{n+1}) \mid \hat{\lambda}] = \frac{B - \mathbb{E}[\ell_{\hat{\lambda}}(X_{n+1}, Y_{n+1}) \mid \hat{\lambda}]}{|\mathcal{I}_2| + 1} \geq 0.$$

By the tower property of expectation, there is

$$\mathbb{E}[\tilde{\alpha}_\ell(\hat{\lambda})] \geq \mathbb{E}[\ell_{\hat{\lambda}}(X_{n+1}, Y_{n+1})].$$

The remaining argument follows the proof of Theorem 3.4 to show an exact validity guarantee for $\hat{\alpha}_\ell(\hat{\lambda})$.

## F. Extension: Inaccurate Solver With Known Suboptimality Gap

The current validity assumes access to exact optimization solutions. However, when only approximate solutions are available, validity can still be guaranteed by incorporating the solver gap into the empirical loss. Denote the suboptimality gap induced by some optimizer solver $\mathcal{A} : \mathcal{Y} \to \mathcal{Z}$ as:

$$G(Y, \mathcal{A}) = f(Y, \mathcal{A}(Y)) - \min_{z \in \mathcal{Z}} f(Y, z).$$

Define $\hat{\ell}_\lambda(x_i, y_i)$ as the solved regret from the solver. We define the solved average regret inflated by the suboptimality gap as:

$$\bar{\ell}_n^{(G)} = \frac{1}{n} \sum_{i=1}^{n} \left[\hat{\ell}_\lambda(x_i, y_i) + G(y_i, \mathcal{A})\right],$$

which is essentially a sample average estimator offset by the optimality gap, we defined the new risk estimator as:

$$\bar{\alpha}_\ell^{(G)}(\lambda) = \frac{n}{n+1} \bar{\ell}_n^{(G)} + \frac{B}{n+1}. \tag{13}$$

It is easy to see that since $\bar{\ell}_n^{(G)} \geq \bar{\ell}_n$, and by applying Theorem 3.4 on $\bar{\alpha}_\ell$, we deduce that:

$$\mathbb{E}[\bar{\alpha}_\ell^{(G)}(\lambda)] \geq \mathbb{E}\left[\bar{\alpha}_\ell(\lambda)\right] \geq \mathbb{E}\left[\ell_\lambda(X_{n+1}, Y_{n+1})\right],$$

which completes the proof for the validity of the new risk estimator defined in (13).

## G. Additional Experiment Details

We focus on four representative optimization paradigms for decision-making problems: $(i)$ linear programming; $(ii)$ the newsvendor problem (single-item inventory); $(iii)$ portfolio optimization (long-only, mean-CVaR proxy); and $(iv)$ the shortest path problem (equivalently, a unit flow problem).

The synthetic data used in our experiments focuses on settings where $X$ is degenerated, so we only have observations of $Y$. As we've mentioned before, this simplification does not limit the generality of our experimental findings, since both the Pareto frontier and its associated performance metrics are inherently unconditioned on $X$.

Our code is written in Python, with core dependencies on the `cvxpy` (Diamond & Boyd, 2016) and `cvxpylayers` (Agrawal et al., 2019) libraries, which are used to solve convex optimization problems. We use the default parameter settings in their libraries across all experiments. Across all settings, the upper bound $B$ is heuristically set to the maximum observed regret for a large number $(1 \times 10^2)$ of simulations. $\lambda$ is varied from zero to the support radius of $Y$.

## G.1. Experimental Setups

In this section, we present the detailed experimental setups for the four optimization problems considered in our study. Across all settings, we specify the uncertainty set as $\mathcal{U}_\lambda = \{\tilde{y} : \|\tilde{y} - \mu\|_\infty \leq \lambda\}$, where $\mu$ denotes the mean of the distribution of the outcome variable $Y$. These optimizations are solved via convex solvers in our implementation.

**Linear Programming**    We consider a two-dimensional linear programming (LP), defined as

$$\min_{z \in \mathbb{R}^2} Y^\top z \quad \text{s.t.} \quad Az \leq b,$$

where $Y$ is a bivariate uniform random variable supported on $[-2.1, -0.1] \times [-2, 0]$. We specify the feasible region as a triacontagon (*a.k.a.* 32-gon) centered at the origin, intersecting the positive orthant. Specifically, the parameters $A \in \mathbb{R}^{34 \times 2}$ and $b \in \mathbb{R}^{34}$ are defined as:

$$A_i = \left( \cos\left(\frac{2\pi}{32} \times i\right), \sin\left(\frac{2\pi}{32} \times i\right) \right), \quad b_i = (1,1), \quad \forall i = 1, \ldots, 32.$$

and the last two rows of $A$ and $b$ are configured to ensure the positivity of $z$ on both dimensions. The robust variant of this problem is defined as

$$\min_{z \in \mathbb{R}^2} \max_{y \in \mathcal{U}_\lambda} y^\top z \quad \text{s.t.} \quad Az \leq b.$$

This robust optimization problem can be equivalently reformulated as another LP

$$\min_{z,t \in \mathbb{R}^2} \mu^\top z + \lambda \mathbf{1}^\top t \quad \text{s.t.} \quad Az \leq b, -t \leq z \leq t, t \geq 0,$$

where $\mathbf{1} = [1,1]^\top$ is an all-ones vector and $t$ serves as an auxiliary variable to linearize the $\ell_\infty$-norm.

**Newsvendor**    We consider a standard single-item inventory (newsvendor) problem, defined as

$$\min_{z \in \mathbb{R}_+} \left[ -p \min(Y, z) + cz - v(z - Y)^+ \right],$$

where $z$ denotes the stocking quantity $Y$ the random demand, $p$ the unit selling price, $c$ the unit procurement cost, and $v$ the unit salvage value for unsold inventory. This objective captures the trade-off between the profit from meeting demand and the loss due to overstocking or understocking. We simulate demands $Y$ as a univariate uniform random variable supported on $[1,3]$, and the parameters are set to $(p, c, v) = (3, 2, 0)$. The robust variant of the newsvendor problem is defined as

$$\min_{z \in \mathbb{R}_+} \max_{y \in \mathcal{U}_\lambda} \left[ -p \min(y, z) + cz - v(z - y)^+ \right].$$

Since both the original problem and the robust problem admit closed-form solutions, we can plug the solutions into the regret expression to get:

$$R_\lambda(X, Y) = p\big(Y - \min\{Y, \mu - \lambda\}\big) - c\big(Y - (\mu - \lambda)\big).$$

**Portfolio Selection**    We study a long-only portfolio selection problem with CVaR regularization. Let $Y \in \mathbb{R}^2$ denote the return scenario for two assets, which is specified as a bivariate uniform random variable supported on $[1,3] \times [1,3]$. Let $z \in \mathbb{R}^2$ be the decision variable. For a given confidence level $\alpha \in (0,1)$ and some risk-aversion parameter $\gamma > 0$, the optimization problem is defined as

$$\min_{z \in \mathbb{R}^2} -Y^\top z - \gamma \operatorname{CVaR}_\alpha(Y^\top z),$$

where the parameters are specified as $(\gamma, \alpha) = (1, 0.95)$. This problem can be reformulated into an explicit convex optimization by introducing an auxiliary VaR variable $t \in \mathbb{R}$ and slack variables $u_i \in \mathbb{R}$:

$$\min_{z,t,u} -Y^\top z + \gamma \left( t + \frac{1}{(1-\alpha)n} \sum_{i=1}^n u_i \right) \quad \text{s.t.} \quad \mathbf{1}^\top z = 1, z \geq 0, u_i \geq -y_i^\top z - t, u_i \geq 0.$$

Here, $y_i$ denotes a separate set of realized set of return scenarios for computing the empirical CVaR. The robust optimization variant can be reformulated by adding an $\ell_1$ penalty on $z$ and replaces $Y$ by $\mu$ to the original optimization problem:

$$\min_{z,t,u} -\mu^\top z + \gamma \left( t + \frac{1}{(1-\alpha)n} \sum_{i=1}^n u_i \right) + \lambda \|z\|_1 \quad \text{s.t.} \quad \mathbf{1}^\top z = 1, z \geq 0, u_i \geq -y_i^\top z - t, u_i \geq 0.$$

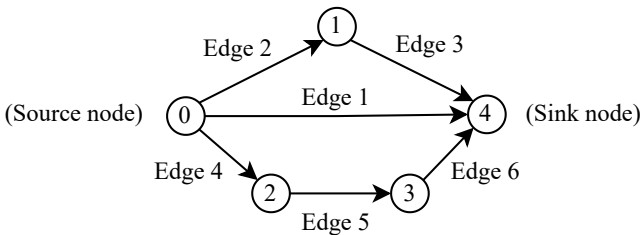

*Figure 5.* An example illustration of a network with three paths considered in our shortest path optimization.

**Shortest Path**    The shortest path problem is formulated on a directed acyclic network with multiple parallel source–sink paths of varying lengths, as illustrated in Figure 5. The objective is to find the least-cost flow that satisfies flow conservation while minimizing travel cost under uncertainty.

Let the network be represented by its node–edge incidence matrix $A \in \mathbb{R}^{n \times m}$ and a supply–demand vector for each node $b \in \mathbb{R}^n$, where $b_0 = 1$ and $b_n = -1$ denote the source and sink nodes, respectively. The decision variable $z \in \mathbb{R}_+^m$ encodes the flow through each edge, and the Gaussian random vector $Y \in \mathbb{R}^m$ specifies the cost on each edge. The nominal (stochastic) shortest path problem is then

$$\min_{z \in \mathbb{R}_+^m} Y^\top z \quad \text{s.t.} \quad Az = b.$$

Since the shortest-path problem is linear, its robust counterpart—and its equivalent convex reformulation—can be written as

$$\min_{z \in \mathbb{R}_+^m} \max_{y \in \mathcal{U}_\lambda} y^\top z \quad \text{s.t.} \quad Az = b. \quad \Longleftrightarrow \quad \min_{z \in \mathbb{R}_+^{n_\text{edge}}} \mu^\top z + \lambda \left( w^\top z \right) \quad \text{s.t.} \quad Az = b,$$

where $w \in \mathbb{R}_+^{n_\text{edge}}$ is a vector of per-edge sensitivity coefficients. Namely, robustness inflates the nominal cost by a weighted $\ell_1$ penalty on flow magnitude.

In our implementation, we specify the network to have three paths, with a gradually decreasing mean of the length random variable $Y$. Please refer to our source code for more detailed specifications.

### G.2. Additional Experiment Results

In this section, we provide additional experimental details and results that were omitted from the main paper.

**Validity-Accuracy Analysis**    We provide additional mathematical details of the evaluation metrics used in the validity-accuracy analysis experiment. Let the number of trials be $T = 20$. For each prespecified robustness parameter $\lambda \in \Lambda$, the validity and accuracy metrics are defined as:

$$\text{Validity} = \frac{1}{T} \sum_{t=1}^{T} \mathbb{1}\{\hat{\alpha}_R(\lambda) \geq \alpha_R(\lambda)\} \cdot \mathbb{1}\{\hat{\alpha}_I(\lambda) \geq \alpha_I(\lambda)\}.$$

$$\text{Accuracy} = -\frac{1}{T} \sum_{t=1}^{T} \sqrt{(\hat{\alpha}_R(\lambda) - \alpha_R(\lambda))^2 + (\hat{\alpha}_I(\lambda) - \alpha_I(\lambda))^2}.$$

In our implementation, these metrics are further averaged over all $\lambda \in \Lambda$, where $\Lambda$ is chosen such that the corresponding uncertainty sets achieve miscoverage rates ranging from $0\%$ to $100\%$. This range is determined by the radius of the support of the distribution of the random outcome $Y$. Finally, the estimators $\hat{\alpha}_\ell(\lambda)$ are computed by averaging over 10 independent replications to obtain smoother estimates.

**Decision Quality Evaluation**    We provide additional details for the setup of the decision quality evaluation experiment. When constructing the certified Pareto frontier estimated from CREME as well as for the baseline methods in estimating $\hat{\lambda}$, we use a total of $n = 10$ calibration data. The optimal solution of the tradeoff is the plug-in of the estimated optimal robustness parameter $\hat{\lambda}$. The estimated tradeoff point from CREME is defined as $\hat{\lambda} = \min_\lambda [w_1 \cdot \hat{\alpha}_I(\lambda) + w_2 \cdot \hat{\alpha}_R(\lambda)]$. Similarly, the optimal tradeoff point from the true Pareto frontier is defined as $\lambda^* = \min_\lambda [w_1 \cdot \alpha_I(\lambda) + w_2 \cdot \alpha_R(\lambda)]$.

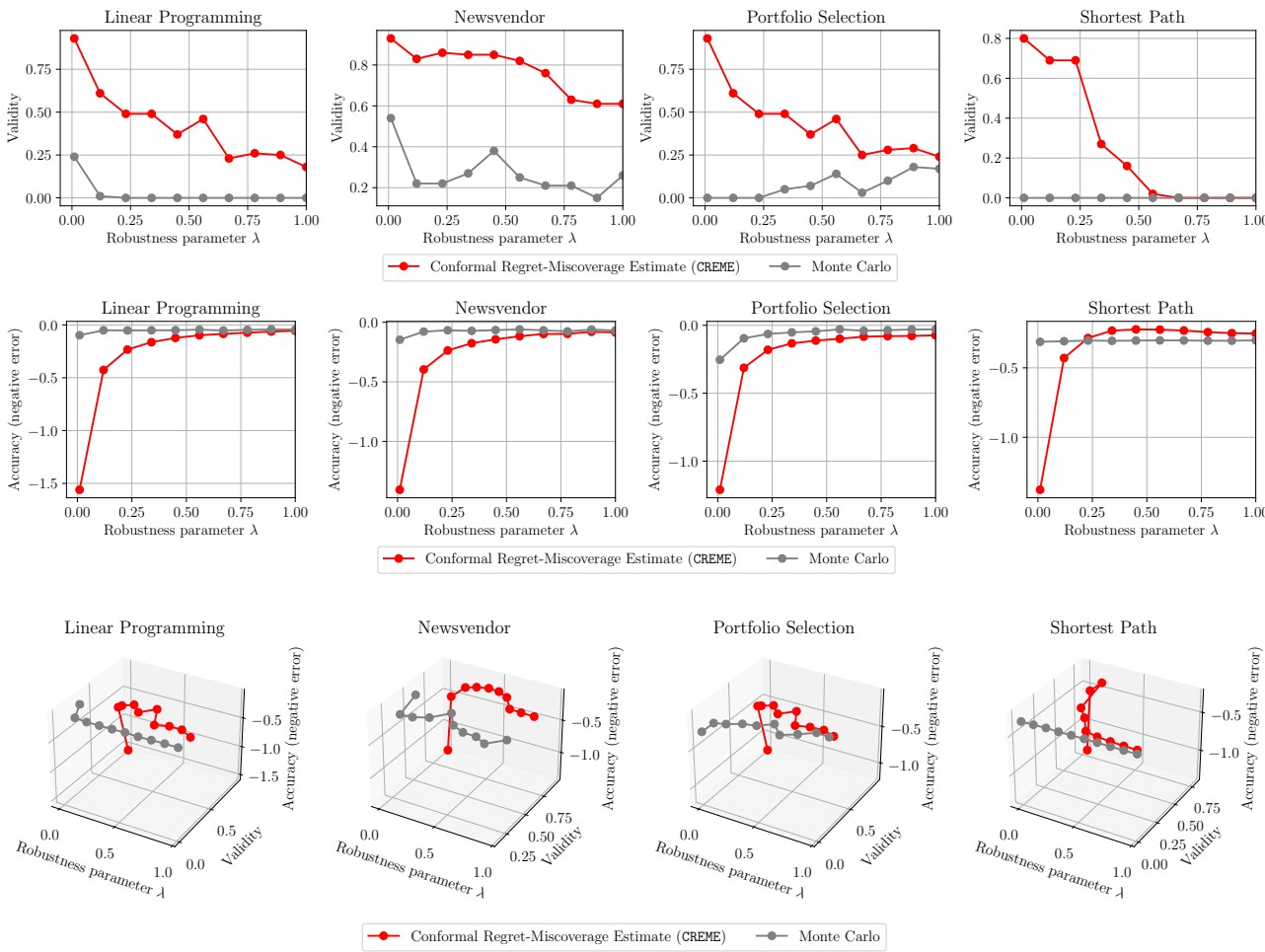

*Figure 6.* Validity-accuracy tradeoff curves under four optimization settings. *First row*: validity versus robustness parameter $\lambda$. *Second row*: accuracy versus robustness parameter $\lambda$. *Third row*: a 3D view of the two metrics and the robustness parameter $\lambda$.

Figure 6 provides a detailed breakdown of the validity–accuracy performance evaluation originally presented in Figure 3 of the main paper. Consistent with the observations discussed in the main text, it is now more clearly visible that as the robustness parameter $\lambda$ (*i.e.*, the radius of the uncertainty set) increases, the validity of CREME decreases approximately linearly, whereas its accuracy improves at an almost exponential rate. These results further substantiate our earlier claim regarding the superior performance of CREME over the naive Monte Carlo estimator.

Figure 7 parallels Figure 4 in the main paper, but presents additional trade-off points attained by different methods under varying specifications of the preference weightings. It can be observed that across different weight configurations, the three baseline methods (parametric quantile, empirical quantile, and conformal prediction) perform well only in the last row, which corresponds to a weighting of $w_1 = 1, w_2 = 0$. This behavior is expected, as this particular weighting places greater emphasis on minimizing miscoverage, which is the primary objective of these baselines. In contrast, CREME consistently identifies near-optimal solutions that lie close to the true Pareto-optimal trade-off points across all weighting specifications. These results further support our main claim that CREME serves as a principled and flexible framework for identifying optimal trade-offs between regret and miscoverage under diverse decision-maker preferences.

**Sensitivity Analysis of Sample Maximum Estimation** The experiment result is shown in Figure 8. In general, the "sample approximated $B$" model produces an estimated Pareto front that is closer to the real Pareto Frontier compared to "Use True $B$". This is because the sample maximum approximation will always be a lower bound to the true $B$, and therefore, the estimated Pareto front will tend to be more radical. This can be an advantage, as the gap between the true and

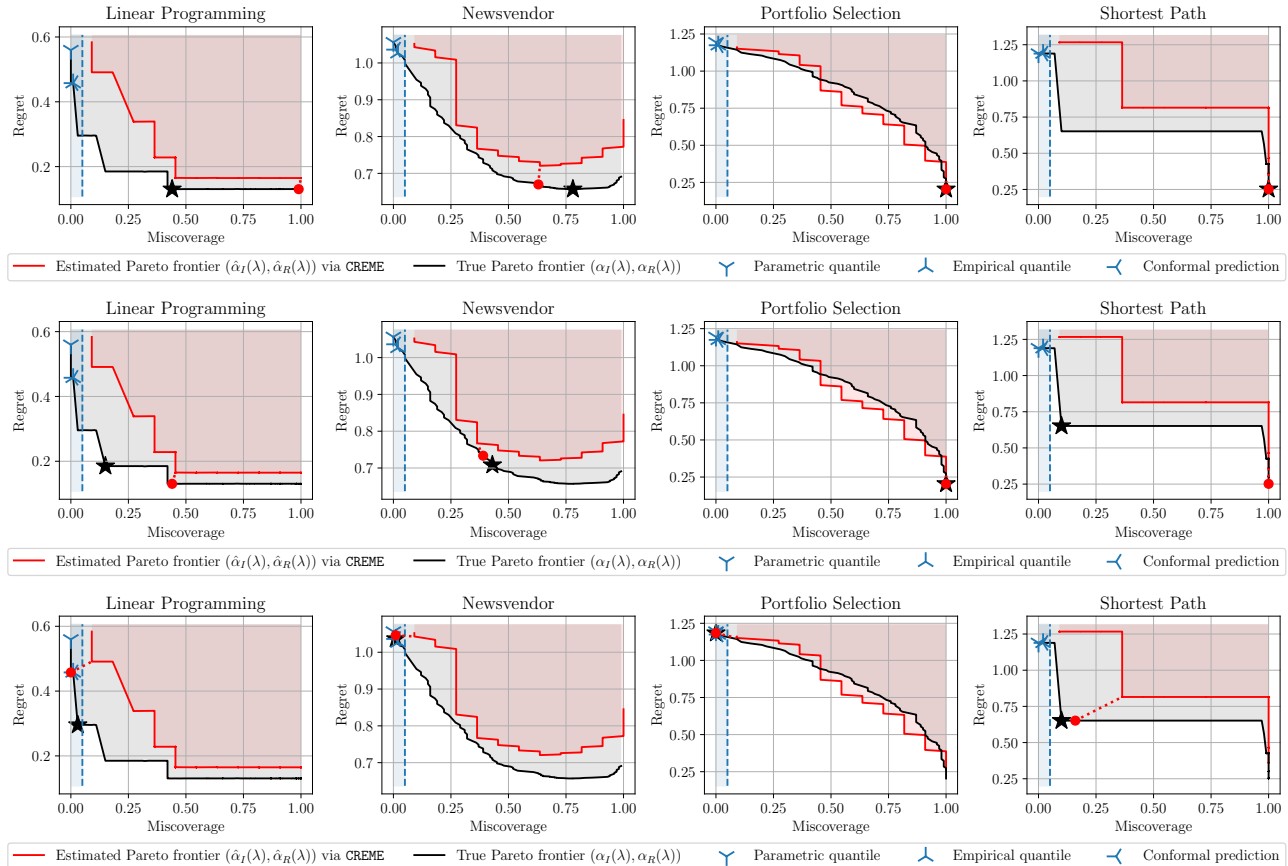

*Figure 7.* Miscoverage–regret tradeoff Pareto frontiers with attained solutions from each model under different preference weights. *First row*: tradeoff points identified using weight $w_1 = 0, w_2 = 1$. *Second row*: tradeoff points identified using weight $w_1 = \sqrt{2}/2, w_2 = \sqrt{2}/2$. *Third row*: tradeoff points identified using weight $w_1 = 1, w_2 = 0$.

estimated Pareto frontier becomes tighter. However, this may also be disadvantageous as it increases the risk of violating validity. For example, in the $n = 30$ setting, it can be seen that the "sample approximated B" model fails to upper bound the real Pareto frontier at the traced red region. This indicates that the "sample approximated $B$" model no longer satisfies Theorem 3.4. As $n$ increases from 10 to 30, both models produce tighter and tighter estimates. Such tightness is also one of the core reasons why "sample approximated $B$" eventually violated validity at $n = 30$.

**Ablation Study on Conformal Correction's Effect on Selected Robustness Parameter**   The experiment result is shown in Figure 9. It can be seen that, across most preference weights and n specifications, the selected robustness parameter is the same for the w/ Correction method and the w/o Correction method. This indicates that, in terms of selected robustness parameters, CREME does not result in a significant difference compared to standard CV robustness tuning approaches. However, a significant difference is observed in terms of the validity of the risk estimate. Across all settings, "w/ Correction" reaches a 100% validity while the "w/o correction" achieves no more than 50%. This indicates that CREME is highly useful for robustness parameter auditing tasks, where credible and reliable risk characterization of the robustness parameter is equally important to its selection.

**Verifying Proposition 3.5 (Finite-Sample Error Bound)**   The experiment result is shown in Figure 10. It can be seen that the empirical MAE (black solid line) and its variations strictly lie below the theoretical upper bound (red dashed line). This serves as empirical evidence supporting that Proposition 3.5 is valid. As $n$ increases, the empirical MAE as well as the theoretical upper bound clearly shrink towards zero. This proves our estimator to be an asymptotically consistent estimator. We note that Proposition 3.5 is not meant to be a tight bound, as the assumption it relies on is minimal (i.e., a bounded assumption), therefore a non-diminishing gap between the empirical and theoretical rate. However, this gap is expected to

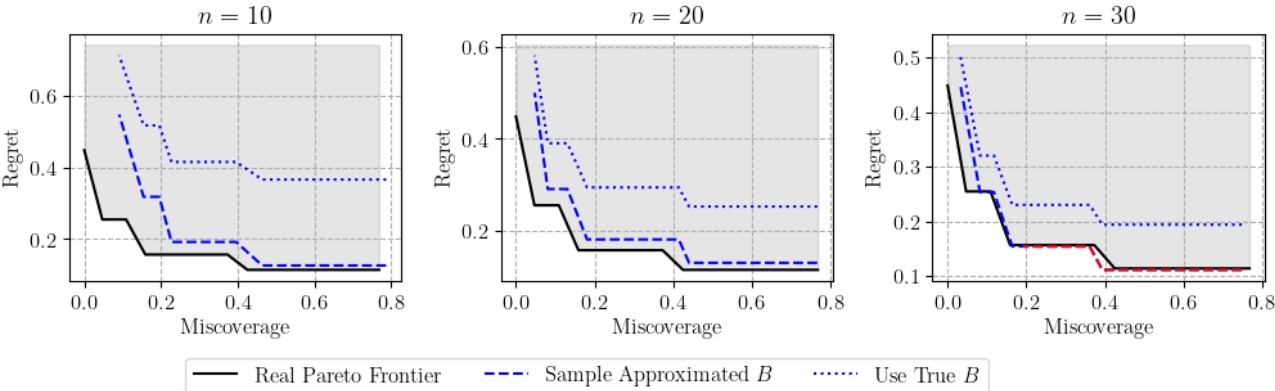

*Figure 8.* Comparison of CREME's estimated Pareto frontier using ground-truth upper bound $B$ versus the sample maximum approximated $B$. The dashed blue line represents the former, and the dotted blue line represents the latter. We vary the total number of data samples used from $n = 10$ to $n = 30$ across three panels. The red dashed line in the right panel indicates failure to satisfy the validity requirement (7) as discussed in Theorem 3.4.

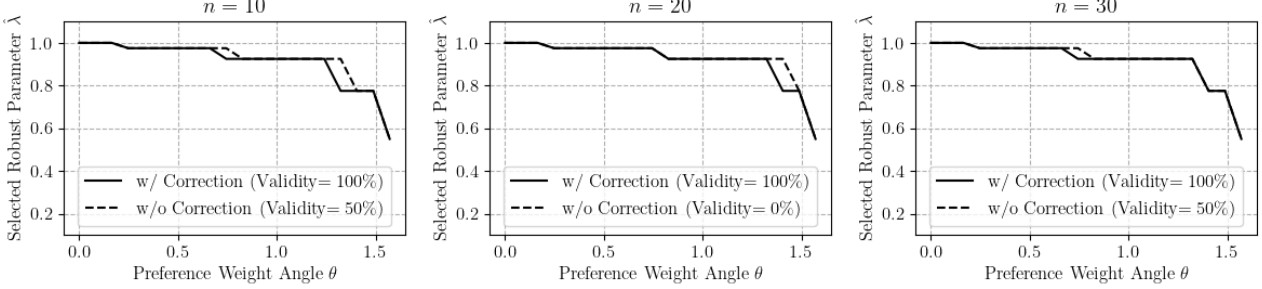

*Figure 9.* Comparison of the selected robustness parameter between conformal correction (*i.e.*, CREME) and without conformal correction (*e.g.*, standard CV robustness tuning) across different numbers of samples $n$. Solid black line indicates "w/ correction" and dashed black line indicates "w/o correction". The decision-maker maximizes a linear preference, where the weight is specified as $(-\sin(\theta), -\cos(\theta))$ with $\theta$ ranging from 0 to $2\pi$. Validity is computed by computing the percentage of trials where (7) in Theorem 3.4 holds empirically.

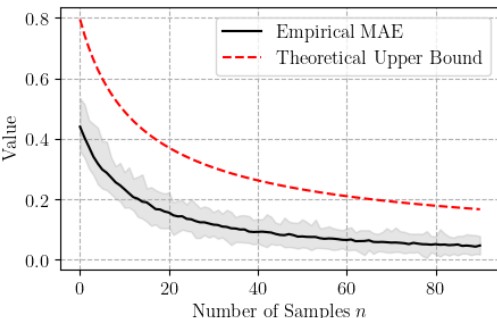

*Figure 10.* Finite-sample mean absolute error (MAE) of the estimated risk and the theoretical bound derived in Proposition 3.5. The red dashed line denotes the theoretical rate, and the black solid line denotes the empirically computed MAE from 100 trials. The shaded area represents the minimum and maximum MAE at the corresponding $n$ across all trials.

close since the theoretical bound would eventually converge to zero.

