# OpenReview forum: "Calibrating Decision Robustness via Inverse Conformal Risk Control"
_ICML.cc/2026/Conference — ICML 2026 regular_

### Official Review · Reviewer_yRdU · 2026-03-05

**Soundness:** 3
**Presentation:** 3
**Significance:** 2
**Originality:** 2
**Overall Recommendation:** 4
**Confidence:** 4

**Summary:**

This paper considers tracing the miscoverage–regret Pareto frontier as the size of the uncertainty set varies. The authors propose distribution-free estimators with finite-sample guarantees. The problem itself is well motivated; however, it would be better if the experimental section could provide more thorough empirical evidence to support the theoretical results.

**Compliance With Llm Reviewing Policy:**

Affirmed.

**Final Justification:**

The authors have provided clear and helpful responses to my questions, along with additional experiments that improve the empirical support and overall clarity of the paper. I appreciate the effort in addressing the concerns raised during the rebuttal. While some limitations remain, I believe they are understandable within the scope of this work. Overall, I view this paper as a technically sound and potentially useful contribution, and therefore I am happy to maintain my original recommendation of weak accept.

**Key Questions For Authors:**

1. In Figure 3, how should we interpret the observation that the validity decreases as ($n$) increases? Could the authors run more trials to empirically verify Theorem 3.4?

2. For the finite-sample error bound in Proposition 3.5, is it possible to derive a result that relies only on the weaker exchangeability assumption rather than the i.i.d. assumption?

3. The error bound in Proposition 3.5 depends on the sample size ($n$). Could the authors empirically verify this bound?

4. Based on the development of the upper-bound estimator, is it also possible to develop a corresponding lower-bound estimator?

5. For the split of the calibration dataset, is there any guideline for choosing the split ratio?

**Limitations:**

yes, in Appendix C

**Strengths And Weaknesses:**

**Strengths**

- The paper is well written, and the authors provide illustrative figures that help readers understand the problem they focus on.
- The problem studied—characterizing the miscoverage–regret Pareto frontier—is relatively novel.

**Weaknesses**

- The proposed risk estimator can be viewed as an upper bound on the risk expectation, but this bound may be quite loose. Assumption 3.2 assumes that the loss is bounded by a constant ($B$), but in practice ($B$) may be large. As a result, the additive adjustment term ($\frac{B}{n+1}$) in Equation (6) can be large when ($n$) is not sufficiently large, making the bound loose. In addition, the calibration dataset needs to be split into two parts, which further reduces the effective sample size in each part and may make the bound even looser.
- Some of the conclusions could benefit from more detailed discussion and interpretation (see the questions below).
- The experimental section is not yet sufficiently comprehensive. More empirical results are needed to validate the theoretical claims and the sample complexity results (see the questions below).
- The results mainly rely on the exchangeability property from conformal prediction, and the conceptual insight seems somewhat limited.

---

> ### Author Rebuttal · Authors · 2026-03-31
>
> We thank the reviewer for the helpful feedback. Please see our rebuttals to the weakness (W) and questions (Q) pointed out by the reviewer below. Cross-references to other responses are often used to meet the rebuttal character limit, and we apologize for any inconvenience.
>
> ## W1
>
> While it is true mathematically that the term $B/(n+1)$ increases as $n$ decreases, this does not imply the risk estimator is "loose". On the empirical side, our experiment results further prove that the estimated Pareto frontier is not "loose". We defer the reviewer to our response to W2 of Reviewer bfey for more details.
>
> The data efficiency concern induced by data splitting can be alleviated by using flexible splitting strategies (see Q5). However, since achieving post-hoc validity is inherently challenging, fully preserving data efficiency is generally impossible without additional assumptions (see our response to Q3 of reviewer qA9b). Therefore, data splitting remains an effective design for our framework.
>
> ## W2
>
> Please refer to our response in Q2.
>
> ## W3
>
> Please refer to our response in Q2.
>
> ## W4
>
> We clarify that exchangeability is standard in conformal prediction and is only used here to ensure the validity of the estimated frontier. CREME imposes no additional structural assumptions beyond this.
>
> This paper introduces a key conceptual insight that broadens both the applicability and impact of robust optimization: selecting a robustness parameter can be reformulated as estimating a certified Pareto frontier. Existing approaches typically rely on heuristics that control a single dimension—cost or risk—often leading to suboptimal performance. In contrast, our framework explicitly characterizes the full certified cost–risk tradeoff via a conformal approach. This enables systematic and reliable auditing of parameter choices, supporting principled decision-making in high-stakes settings. We view this conceptual insight as a central contribution of the work.
>
> ## Q1
>
> First of all, we clarify that the validity metric used in the experiment (defined in lines 359-361) is "stricter" than the guarantee established in Theorem 3.4. It approximates violation probability at a pointwise level, akin to family-wise error control, whereas Theorem 3.4 guarantees validity in expectation. Therefore, in Figure 3, the observation that validity decreases as $n$ increases is expected: as the correction term shrinks, noise in the data makes it more likely that at least one point violates the frontier, decreasing this validity metric.
>
> To avoid confusion, we redefined the validity metric to directly measure whether the guarantee in Theorem 3.4 is satisfied and renamed the original validity metric to "violation frequency". Additionally, to further empirically validate Theorem 3.4, we increased the number of trials from $T = 20$ to $T = 100$. We will provide these results as additional replies to the review due to the rebuttal character limits.
>
> ## Q2
>
> Under pure exchangeability alone, a bound similar to Prop. 3.5 is generally not available. The issue is that exchangeability allows strong dependency, which could result in arbitrarily loose bounds.
>
> For example, let $\theta \sim \mathrm{Bernoulli}(1/2)$ and $\ell_\lambda(X_i,Y_i)=B\theta$ for all $i$. The sequence is exchangeable but perfectly dependent. Its empirical mean is always $0$ or $B$ and does not concentrate around $B/2$ as the sample size grows. Thus, the estimation error remains order-$B$ with constant probability, which differs from the asymptotically consistent bound established in Prop. 3.5.
>
> ## Q3
>
> Due to the rebuttal character limits, we will provide these results as additional replies to the review.
>
> ## Q4
>
> A lower-bounding estimator can be easily derived by reversing the conformal procedure in Eq. (5) as:
> $$
> \sup_{\alpha \in (0, B)} \left\[
> \alpha : \bar \ell_n(\lambda) \leq \frac{\lfloor (n + 1) \alpha \rfloor}{n}
> \right\].
> $$
> It can be further simplified and approximated by removing the ceiling operator in this expression, yielding the closed-form estimator $\tilde \alpha(\lambda) = \frac{n}{n + 1} \bar \ell_n(\lambda)$.
>
> ## Q5
>
> In general, we recommend the user default to a half-to-half split ratio to maintain a balanced performance between tightness and optimality. This is also used across all of our experiment settings.
>
> In data-scarce settings, for higher data-efficiency, we recommend: (i) Skip the second split and use the full dataset to estimate the Pareto frontier. This restores full data efficiency and still preserves the validity of the estimation, but requires it to be used without any post-hoc selection, as bias may be introduced; Or (ii) allocate uneven splits by assigning a larger pre-hoc dataset and a smaller post-hoc dataset. This can yield a tighter estimated Pareto frontier compared to an even split, but may result in a less optimal choice of the robustness parameter.

---

> > ### Author Rebuttal · Reviewer_yRdU · 2026-04-03
> >
> > Thank you for your detailed response. The clarification is clear, but additional experiments on more complicated dataset are expected, especially for W1 and W3. W1 primarily comes from the inherent limitation of the conformal risk control technique, so I would expect further development to break this limiation, or experiments to discuss the impact of the bounded assumption in real-world applications.

---

> > > ### Author Response · Authors · 2026-04-03
> > >
> > > We thank the reviewer for the response. Please access our additional experiment results through this anonymous website link: https://sites.google.com/view/creme-rebuttal/.
> > >
> > > Regarding the reviewer's concern about the boundedness assumption on $B$ (W1), we additionally note that although the assumption is important for deriving Theorem 3.4, it is not essential for the practical implementation of CREME.
> > > As pointed out in our response to W1 of reviewer qA9b, $B$ can be approximated via the sample maximum $\max_i \ell_\lambda(x_i,y_i)$ in practice, which extends CREME to many settings where $B$ is unknown and/or does not exist. Please use the link to view our evaluation result comparing the sample maximum estimator of $B$ with its ground truth across different sample sizes.
> > >
> > > For W3, we have conducted additional experiments to empirically validate both Theorem 3.4 and Proposition 3.5 across different optimization settings and sample sizes $n$, which aligns well with the theories. These results are also uploaded to the website.
> > >
> > > We hope we have fully addressed your concerns. If not, we are happy to continue the discussion on how we can further improve the current manuscript and/or include additional experiment results. Thank you!

---

### Official Review · Reviewer_qA9b · 2026-03-11

**Soundness:** 3
**Presentation:** 2
**Significance:** 2
**Originality:** 2
**Overall Recommendation:** 4
**Confidence:** 4

**Summary:**

The paper introduces Conformal REgret Miscoverage Estimate (CREME), an "inverse" conformal risk control framework designed to evaluate and select robustness parameters in robust optimization (RO). Rather than fixing a risk tolerance a priori and searching for an uncertainty set, the authors reverse the paradigm: they take a parameterized uncertainty set (indexed by $\lambda$) as a given and provide rigorous upper bounds on both the resulting miscoverage and expected regret. By doing so, the method establishes a certified Pareto frontier that allows decision-makers to transparently balance protection against conservativeness. The authors prove finite-sample validity for these bounds and introduce a data-splitting approach to maintain validity even when the robustness level is selected post-hoc. The empirical performance is validated on four classical optimization problems (Linear Programming, Newsvendor, Portfolio Selection, and Shortest Path).

**Compliance With Llm Reviewing Policy:**

Affirmed.

**Final Justification:**

The answer to Q1 and Q2 make sense. So I increase my score to 4.

**Key Questions For Authors:**

1. CREME fundamentally relies on the exchangeability of the calibration and test data (Assumption 3.1). However, many of the motivating examples, such as financial portfolio management and weather-dependent grid dispatch, are time-series problems subject to severe distribution shifts. Are there pathways to extend Inverse CRC using non-exchangeable conformal methods (e.g., adaptive or weighted conformal prediction)?
2. Because evaluating $\min_{z \in \mathcal{Z}} f(Y, z)$  might be intractable for complex problems, what happens to your validity guarantees if we can only compute an approximate oracle solution (e.g., finding a solution with a known optimality gap)? Can the CREME framework mathematically incorporate suboptimality gaps into the regret bound?
3. While data splitting restores post-hoc validity, it halves the effective calibration sample size. Is it possible to solve the problem by leveraging full conformal techniques or other methods?

**Limitations:**

The authors are encouraged to address the identified weaknesses and questions to improve the paper.

**Strengths And Weaknesses:**

**Strengths:**

 1. The shift from generative Conformal Risk Control (CRC) to "inverse" diagnostic CRC is highly practical. Practitioners frequently struggle with setting RO uncertainty sets without being overly conservative; providing a certified Pareto frontier to diagnose these trade-offs directly addresses a significant operational pain point.
 2. The framework offers distribution-free, finite-sample guarantees and establishes a rigorous finite-sample error bound
 3. The experiments are well-organized and the supplementary results are comprehensive.

**Weaknesses:**

 1. The framework requires a known and finite upper bound $B$ for the loss function, which is a limitation inherited from standard CRC. However, in unbounded outcome spaces or complex non-linear optimization tasks, tightly bounding the maximum possible regret analytically may be difficult. A sensitive analysis on $B$ would have stengthen the contribution.

2. While data splitting restores post-hoc validity, it halves the effective calibration sample size. In data-scarce environments, this could lead to wider bounds and reduced accuracy. Table 1 shows that accuracy drops noticeably when $n$ is small (e.g., $n=10$), meaning the data splitting cost is non-trivial.

---

> ### Author Rebuttal · Authors · 2026-03-31
>
> We thank the reviewer for the helpful feedback. Please see our rebuttals to the weakness (W) and questions (Q) pointed out by the reviewer below.
>
> ## W1
>
> The bounded loss assumption, inherited from conformal risk control, is used to derive the validity guarantees (e.g., Theorem 1). However, we note that in practical implementations, exact knowledge of $B$ is not required if exact validity is not needed. For example, $B$ can be approximated via the sample maximum $\max_i \ell_\lambda(x_i,y_i)$. We further conduct a sensitivity analysis of $B$, comparing the sample maximum estimator of $B$ with its ground truth across different sample sizes. Due to the rebuttal character limits, we will provide these results as additional replies to the review.
>
> ## W2
>
> While data splitting reduces efficiency, the split need not be equal. In data-scarce settings, one may: (i) skip the second split and use the full dataset, which maintains validity but disallows post-hoc selection; or (ii) use uneven splits, which improves frontier tightness at the cost of potential suboptimal parameter choice. Additionally, we emphasize that post-hoc validity is inherently difficult to achieve. To the best of our knowledge, it typically requires either reduced data efficiency or stronger assumptions. Please see our response to Q3 for further discussion.
>
> ## Q1
>
> We clarify that exchangeability is standard in conformal prediction and is only used here to ensure the validity of the estimated frontier. CREME imposes no additional structural assumptions beyond this.
>
> Adapting CREME to non-exchangeable settings (e.g., time series) is a promising direction for future research. A modified version of the CREME risk estimator inspired by the weighted conformal prediction [1] takes the form:
> $$
> \tilde \alpha_\ell (\lambda) = \frac{n}{n + 1} \bar \ell_n^{(w)}(\lambda) + \frac{B}{n + 1},
> $$
> where $\bar \ell_n^{(w)}(\lambda)$ is defined as the weighted sample average:
> $$
> \bar \ell_n^{(w)}(\lambda) = \frac{1}{n} \sum_{i=1}^n \frac{w(x_i)}{w(x_i) + w(x)} \ell_\lambda(x_i, y_i),
> $$
> and $w(x)$ is defined as the likelihood-ratio between the original distribution and the shifted distribution. Further justifications of its form in the revised manuscript.
>
> > [1] Tibshirani, R. J., Foygel Barber, R., Candes, E., & Ramdas, A. (2019). Conformal prediction under covariate shift. Advances in neural information processing systems, 32.
>
> ## Q2
>
> The current validity assumes access to exact optimization solutions. However, when only approximate solutions are available, validity can still be guaranteed by incorporating the solver gap into the empirical loss. Denote the suboptimality gap induced by some optimizer solver $\mathcal{A}: \mathcal{Y} \to \mathcal{Z}$ as
> $$
> G(Y, \mathcal{A}) = f(Y, \mathcal{A}(Y)) - \min_{z \in \mathcal{Z}} f(Y, z).
> $$
> Define the term:
> $$
> \bar \ell_n^{(G)}(\lambda) = \frac{1}{n} \sum_{i=1}^n \left[ \ell_\lambda(x_i, y_i) + G(y_i, \mathcal{A})\right],
> $$
> which is essentially a sample average estimator offset by the optimality gap, we defined the new risk estimator as:
> $$
> \tilde \alpha_\ell (\lambda) = \frac{n}{n + 1} \bar \ell_n^{(G)}(\lambda) + \frac{B}{n + 1}.
> $$
> This modification fully restores the validity guarantees of the algorithm, which can be easily proven by noticing how the added suboptimality gap offsets the solver errors corresponding. A full documentation of this procedure and associated proof is included in the revised manuscript.
>
> ## Q3
>
> Full conformal techniques cannot handle post-hoc selection bias. However, we discuss two popular methods in the literature that may be adapted in our setting to handle this issue:
>
> - Selective inference [2] explicitly accounts for the data-dependent selection step by conditioning on the selection event (e.g., the chosen $\hat{\lambda}$), and then explicitly deriving a selection-aware correction term to replace the $B/(n+1)$. This approach preserves full data efficiency, but requires strong assumptions on the data distribution and the selection mechanism that is beyond the consideration of the current framework.
>
> - Algorithm stability [3] aims to “de-correlate” the estimator from the selected $\hat{\lambda}$ by injecting random noise into the post-hoc selection procedure. This approach avoids explicit data splitting, but may result in selecting a suboptimal $\hat \lambda$ as it is chosen based on a deliberately perturbed rather than the original objective.
>
> Overall, while both of these procedure preserving data efficiency, they each come at other costs. Therefore, data splitting remains justified for its simplicity and effectiveness in addressing post-hoc validity.
>
> > [2] Berk, R., Brown, L., Buja, A., Zhang, K., & Zhao, L. (2013). Valid post-selection inference. The Annals of Statistics, 802-837.
>
> > [3] Zrnic, T., & Jordan, M. I. (2023). Post-selection inference via algorithmic stability. The Annals of Statistics, 51(4), 1666-1691.

---

> > ### Author Rebuttal · Reviewer_qA9b · 2026-04-03
> >
> > Thanks for the detailed feedback. Most of my concerns have been addressed and I will increase my score. Good luck!

---

> > > ### Author Response · Authors · 2026-04-03
> > >
> > > Thank you for the response. We are glad that we have addressed your concerns, and we truly appreciate your positive assessment.

---

### Official Review · Reviewer_bfey · 2026-03-13

**Soundness:** 3
**Presentation:** 3
**Significance:** 2
**Originality:** 3
**Overall Recommendation:** 4
**Confidence:** 4

**Summary:**

This paper studies the problem of calibrating robustness levels in robust optimization and predict‑then‑optimize pipelines. Unlike the case of standard conformal risk control, where the decision maker fixes a target miscoverage level a priori, the authors propose an inverse conformal risk control framework that estimates finite sample upper bounds on miscoverage and regret for a family of robust policies indexed by a robustness parameter. By evaluating these bounds across different robustness levels, the method traces a certified miscoverage–regret Pareto frontier, allowing the decision maker to select robustness post hoc.

**Compliance With Llm Reviewing Policy:**

Affirmed.

**Final Justification:**

The rebuttal has addressed the concerns I have raised. Upon going through the additional results and the discussions of the other reviewers, I am increasing my score to weak accept.

**Key Questions For Authors:**

1. How would one go about to tighten the estimator beyond the B/(n+1) correction, while retaining finite‑sample validity?

2. How should one interpret or apply the framework when majorant consistency does not hold (e.g., nonconvex decision rules, learned policies)? Is the Pareto interpretation still meaningful?

3. In practice, how sensitive are robustness selections to the looseness of the bounds? Do the certified frontiers change decisions compared to empirical estimates?

4. How does the proposed framework compare to data splitting or cross validation based robustness tuning without conformal correction?

**Limitations:**

Yes

**Strengths And Weaknesses:**

Strengths:

1. The paper correctly identifies that selecting robustness parameters in robust optimization is typically ad hoc and that there is a need for a better framework for practical use.

2. Using the exchangeability assumptions and appropriate analyze the loss of validity under post‑hoc parameter selection, the authors propose a data‑splitting fix that is intuitive and implementation friendly.

3. The framework applies to any bounded losses like miscoverage, regret etc., and is not tied to a specific optimization problem or uncertainty set.

4. The paper is clearly presented, with explicit assumptions, illustrative figures, and a complete theoretical analysis including counterexamples.

Weaknesses:

1. The estimator is essentially a conservative conformal upper bound on expected loss, combined with standard enumeration over robustness levels. While the auditing perspective is useful, the technical contribution largely recombines existing ideas from conformal risk control, inverse conformal prediction, and Pareto analysis without introducing new statistical mechanisms or tighter bounds.

2. The key correction term scales as B/(n+1), which dominates in small and moderate sample regimes. As a result, the certified Pareto frontiers are often loose, and it is unclear how actionable the method is in realistic data‑scarce settings where robustness calibration is most needed.

3. The interpretation of the estimated curve as a true Pareto frontier depends critically on monotonic regret assumptions. While standard in classical convex RO, this assumption is week outside those settings and can potentially limit applicability to modern ML‑based decision pipelines.

---

> ### Author Rebuttal · Authors · 2026-03-31
>
> We thank the reviewer for the helpful feedback. Please see our rebuttals to the weakness (W) and questions (Q) pointed out by the reviewer below.
>
> ## W1
>
> Establishing a conservative conformal upper bound on expected loss is nontrivial. While CRC selects $\lambda$ to satisfy a loss bound, the inverse problem—conformalizing loss for a given $\lambda$—has not been studied. We are the first to formulate and solve this problem via a principled conformal approach. In the meantime, the technical contribution of this work goes beyond recombining existing ideas from CRC and inverse conformal prediction. We derive Eq. (5) by inverting CRC and optimizing over $\alpha$ instead of $\lambda$, and further obtain a closed-form approximation enabling efficient implementation. These results are new to the conformal literature.
>
> ## W2
>
> While it is mathematically true that $B/(n+1)$ increases as $n$ decreases, this does not imply the risk estimator is "loose".
> Theoretically, $B/(n+1)$ is essentially the tightest correction ensuring validity under our assumptions, and is standard in conformal methods (e.g., CRC). Therefore, under a small/moderate sample regime, $B/(n+1)$ reflects the least amount of slackness to guard against the worst-case scenario, hence is the inherent price that needs to be paid to achieve validity.
>
> Our experiment results also prove that the estimated Pareto frontier is not "loose". In Table 1, the reported average gap between the estimated and true Pareto frontier is around $0.1$. Since we set $B \geq 1$, the estimation gap is $\approx 10$%. Such an error is minimal and acceptable in high-stakes settings where decision-makers want safe and reliable robust parameter choices, as violation of conservativeness could be more dangerous. Additionally, the gap is expected to be even smaller as the data volume could easily reach beyond, e.g., $n = 100$, in real implementations where large amounts of data are available, while our experiments use only $n \leq 30$.
>
> Additionally, our experiments demonstrate that the method can provide meaningful guidance to parameter selection in data-scarce settings. Figure 4 shows that under $n=20$, the selected tradeoff points (solid red/green) closely align with the ground-truth optimum (black star), outperforming baselines. Together with Table 1, these results provide strong evidence of the method's practical effectiveness in data-scarce regimes.
>
> ## W3
>
> We clarify that the monotonicity of $\ell_\lambda$ in $\lambda$ is not a formal assumption and was stated only for presentation. Violating it does not affect the interpretation of the estimated Pareto frontier. We will remove this description in the revised manuscript to avoid confusion.
>
> If the reviewer is referring to the majorant consistency assumption, we clarify that it is also not essential to the Pareto analysis. It only ensures that the traced trajectory forms a well-defined frontier, rather than an irregular shape such as a swirl or blob. When this assumption is violated, one can manually prune out a well-defined Pareto frontier by removing the robustness parameters $\lambda \in \Lambda$ that correspond to non-dominating points on the trajectory. This has already been described in Step 11 of Algorithm 1 in the original manuscript. Thus, the framework also covers settings where this assumption fails, such as the modern ML-based decision pipelines as mentioned by the reviewers.
>
> ## Q1
>
> Tightening the correction term $B/(n+1)$ while retaining finite-sample validity is generally impossible without imposing additional assumptions. If that is acceptable, there are two potential approaches adaptable to our setting: [1] introduces a DKWM-based correction for conformal prediction that scales as $\mathcal{O}\left(\sqrt{\frac{\log(1/\delta)}{n}}\right)$, providing a guarantee that holds with high probability at level $1-\delta$, but assumes i.i.d. data. On the other hand, [2] studies a binomial-law–based characterization of empirical coverage in conformal prediction, which yields corrections scaling as $\mathcal{O}(n^{-1})$, but relies on expressing coverage via indicator variables, which applies to miscoverage but not regret.
>
> > [1] Vovk, V. (2012, November). Conditional validity of inductive conformal predictors. In Asian conference on machine learning (pp. 475-490). PMLR.
>
> > [2] Marques, F., & Paulo, C. (2025). Universal distribution of the empirical coverage in split conformal prediction. Statistics & Probability Letters, 219(C).
>
> ## Q2
>
> Please see our response to W3.
>
> ## Q3
>
> Please see our response to W2.
>
> ## Q4
>
> We conduct an additional experiment comparing the robustness parameter selected from the estimated Pareto frontier with conformal correction (w/ Correction, ours) against that obtained via standard robustness tuning based on the estimated Pareto frontier without conformal correction (w/o Correction). Due to the rebuttal character limits, we will provide these results as additional replies to the review.

---

> > ### Author Rebuttal · Reviewer_bfey · 2026-04-03
> >
> > The authors addressed my comments, and I don't have further questions.

---

> > > ### Author Response · Authors · 2026-04-03
> > >
> > > Thank you for your response, and we are glad that we have adequately addressed your concerns. If you believe it is warranted, we would greatly appreciate it if you could consider updating your score accordingly. Please also feel free to check out some of our additional experiment results through this anonymous website https://sites.google.com/view/creme-rebuttal/

---

### Decision · Program_Chairs · 2026-04-30

**Decision:**

Accept (regular)

**Comment:**

All reviewers agree that the paper makes a solid contribution. The reviewers suggest further clarification of the assumptions and the potential additional sample complexity of the proposed method. Especially, for the latter one, I believe it is a valid concern and limitation that we need more data for calibrating decision robustness. I suggest the authors incorporate these suggestions and make the promised changes in their revision.